# Proteome activity landscapes of tumor cell lines determine drug responses

Martin Frejno [1,11], Chen Meng [1,2,11], Benjamin Ruprecht [1,3,10,11], Thomas Oellerich [4,5,6], Sebastian Scheich [4], Karin Kleigrewe [2], Enken Drecoll [7], Patroklos Samaras [1], Alexander Hogrebe [1], Dominic Helm [1], Julia Mergner [1], Jana Zecha [1], Stephanie Heinzlmeir [1], Mathias Wilhelm [1], Julia Dorn [8], Hans-Michael Kvasnicka [9], Hubert Serve [4,6], Wilko Weichert [6,7] & Bernhard Kuster [1,2,3,5 ✉]

Integrated analysis of genomes, transcriptomes, proteomes and drug responses of cancer cell lines (CCLs) is an emerging approach to uncover molecular mechanisms of drug action. We extend this paradigm to measuring proteome activity landscapes by acquiring and integrating quantitative data for 10,000 proteins and 55,000 phosphorylation sites (p-sites) from 125 CCLs. These data are used to contextualize proteins and p-sites and predict drug sensitivity. For example, we find that Progesterone Receptor (PGR) phosphorylation is associated with sensitivity to drugs modulating estrogen signaling such as Raloxifene. We also demonstrate that Adenylate kinase isoenzyme 1 (AK1) inactivates antimetabolites like Cytarabine. Consequently, high AK1 levels correlate with poor survival of Cytarabine-treated acute myeloid leukemia patients, qualifying AK1 as a patient stratification marker and possibly as a drug target. We provide an interactive web application termed ATLANTiC (http://atlantic. proteomics.wzw.tum.de), which enables the community to explore the thousands of novel functional associations generated by this work.

[1] Chair of Proteomics and Bioanalytics, Technical University of Munich, Emil-Erlenmeyer-Forum 5, 85354 Freising, Germany. [2] Bavarian Center for Biomolecular Mass Spectrometry, Technical University of Munich, Gregor-Mendel-Strasse 4, 85354 Freising, Germany. [3] Center for Integrated Protein Science Munich (CIPSM), Butenandtstr. 5-13, 81377 Munich, Germany. [4] Department of Medicine, Hematology/Oncology, Goethe-University, Theodor-Stern-Kai 7, 60590 Frankfurt, Germany. [5] Lymphoid Malignancies Branch, National Cancer Institute, National Institutes of Health, Bethesda, MD 20892-1374, USA. [6] German Cancer Consortium (DKTK), Im Neuenheimer Feld 280, 69120 Heidelberg, Germany. [7] Institute of Pathology, Technical University of Munich, Trogerstr. 18, 81675 Munich, Germany. [8] Klinikum rechts der Isar, Technical University of Munich, Ismaninger Straße 22, 81675 Munich, Germany. [9] Institute of Pathology, Goethe University, Theodor-Stern-Kai 7, 60590 Frankfurt, Germany. [10] Present address: Chemical Biology, Merck Research Laboratories, 33 Avenue Louis Pasteur, Boston, MA 02115, USA. [11] These authors contributed equally: Martin Frejno, Chen Meng, Benjamin Ruprecht. ✉email: kuster@tum.de

ntegration of genomic, transcriptomic, and proteomic profiles of tumor cell lines with phenotypic drug response data has extended our understanding of tumor biology and helped to delineate the mechanisms of action (MoA) of several drugs[1,2]. However, post-translational modifications (PTMs) have been rarely investigated systematically in the context of drug sensitivity[3], even though it is well-established that proteome activity regulated by e.g., dynamic phosphorylation plays a major role in cancer initiation, progression, and response to drugs. Consequently, many kinase inhibitors that target phosphorylation-regulated signaling pathways have been developed as cancer drugs, and some have transformed the clinical management of several cancer entities[4,5].

Here, we profile the baseline phosphoproteomes of the NCI60 and CRC65 cell line panels (Fig. 1a) to an overall depth of >55,000 p-sites using a consistent and reproducible mass spectrometry workflow (Fig. 1b), in order to further improve our understanding of the molecular mechanisms of action (MoA) of cancer drugs and how the signaling repertoire of cancer cells affects drug response. In addition, we reacquire data on the proteomes of the NCI60 cell line panel[6] to a depth of >10,000 proteins. The newly generated proteomic and phosphoproteomic data is integrated with our previously published CRC65 data[7], as well as with published phenotypic drug sensitivity (~900 drugs) and published molecular drug target selectivity information (224 drugs) from our laboratory[8] (Fig. 1a). Based on these data, we

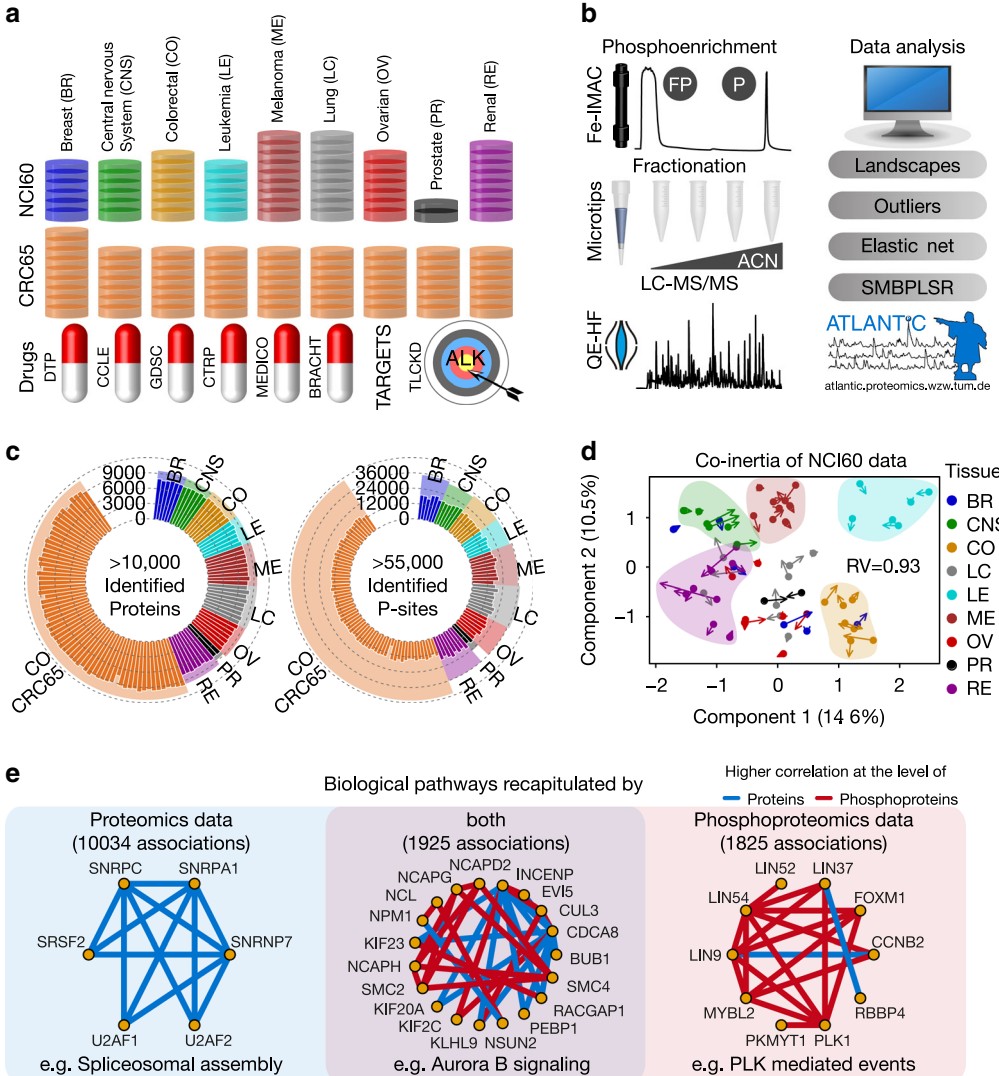

**Fig. 1 Proteome and phosphoproteome profiling of 125 tumor cell lines. a** Overview of the datasets included in this study, covering the NCI60 (*n* = 60 cell lines) and CRC65 cell line panels (*n* = 65 cell lines), six public drug-sensitivity datasets and a dataset containing drug targets of clinical kinase inhibitors (see Supplementary Methods for details on all datasets). Cell lines are colored by tissue of origin. This color scheme is consistent across figures. **b** Schematic representation of the biochemical and data analysis workflows. FP and P denote full proteome and phosphoproteome, respectively. The results can be explored interactively on the ATLANTiC website. (ACN = acetonitrile, SMBPLSR = sparse multiblock partial least squares regression) **c** Circular bar plot showing the number of identified proteins and p-sites per cell line (*n* = 125 cell lines). Shaded areas indicate the sum of the proteins and p-sites within each tumor entity. We identified a grand total of >10,000 protein groups and >55,000 p-sites. **d** Multiple co-inertia analysis of phospho- and full proteome data of the NCI60 panel showing the first two components (*n* = 59 cell lines). Bases and ends of arrows represent the full proteome and phosphoproteome data of a given cell line, respectively. Short arrows indicate a good correlation between phospho- and full proteome. The RV coefficient quantifies the correlation of two matrices analogous to the Pearson correlation coefficient. A RV close to 1 indicates high correlation. **e** Representative pathways significantly (Fisher's exact test; Benjamini–Hochberg corrected *P* < 0.05) enriched in functional associations recapitulated by our proteomics data (left panel), phosphoproteomics data (right panel) or both (middle panel). Source data are provided as a Source Data file.

compute activity landscapes of tumor cell lines and compare both outlier and correlation-based drug response markers in relative terms among cell lines. The raw mass spectrometric data and processed tandem mass spectra are accessible for further exploration via the PRIDE[9] repository and ProteomicsDB[10], respectively. In addition, all of the data can be explored using the interactive web application ATLANTiC (http://atlantic.proteomics.wzw.tum.de), which allows users to visualize activity landscapes and protein/p-site abundance information across the entire dataset, to query the results of all drug response modeling analyses and to deconvolute the target space of kinase inhibitors in specific cell lines. We envision that ATLANTiC will enable the research community to perform many additional investigations based on the tens of thousands of observations in this study, only few of which can be highlighted in this report. For example, we reveal that PGR phosphorylation is correlated with sensitivity to endocrine therapy in PGR-positive breast cancer. We also show that AK1 is capable of inactivating antimetabolites like Cytarabine and that AK1 protein levels correlate with poor survival of Cytarabine-treated acute myeloid leukemia patients. This qualifies AK1 as a patient stratification marker and possibly as a drug target.

## Results

**Comparison of proteomics and phosphoproteomics data.** Figure 1c gives an overview of the depth of this highly consistent dataset (Supplementary Fig. 1A–E, Supplementary Data 1–3, Supplementary Methods), which is the basis for all integrative analyses presented herein and available through our ATLANTiC web application, such as the deconvolution of the target space of kinase inhibitors in specific cell lines (Supplementary Fig. 1F, G).

Focusing on differences and similarities between proteomics and phosphoproteomics data, multiple co-inertia analysis[11] based on the proteins and p-sites of the NCI60 panel confirms the previous notion[6] that cell lines from certain tumor entities are molecularly similar to each other (e.g., leukemia), whereas other entities are highly heterogeneous (e.g., breast cancer; Fig. 1d, Supplementary Methods). The analysis also reveals that baseline p-site abundance generally follows protein abundance (short arrows; RV coefficient 0.93 for NCI60 and 0.90 for CRC65 data; Supplementary Fig. 1H), indicating that phosphorylation has many roles in basic cellular homeostasis beyond its dynamic regulation of cellular signaling. To investigate this in more detail, we asked which biological pathways recorded in pathway databases are recapitulated at the level of proteins, phosphoproteins or both in all of the cell lines. To be able to compare the phosphoproteomics data with public pathway databases, which only contain information at the protein level, we first summed up all phosphopeptide intensities for each cell line and protein group to yield phosphoprotein intensities (Supplementary Methods) similar to what was done previously[12]. The authors point out that this is a strong simplification because phosphorylation of different sites on the same protein may be regulated by different kinases and phosphatases and can result in different effects on cellular signaling (e.g., activating versus inactivating p-sites). We then focused on functional associations between any two proteins recorded in pathway databases and compared the abundance correlation of these two proteins across the NCI60 and CRC65 cell line panels at the protein and phosphoprotein level (Supplementary Methods). Interestingly, only 32 and 51% of the functional associations recapitulated at the phosphoprotein level are also recapitulated at the protein level in the NCI60 and CRC65 dataset, respectively (Supplementary Fig. 2). Therefore, we performed separate analyses at the protein, phosphoprotein and p-site level throughout this manuscript. Enrichment analysis

of functional associations (Supplementary Data 4) suggests that biological pathways recapitulated at the protein level are enriched in basic cellular functions such as spliceosomal assembly. In contrast, functional associations recapitulated at the phosphoprotein level are enriched in dynamic processes such as PLK-mediated events in the cell cycle. Functional associations recapitulated at both levels include the cell-cycle-associated Aurora B signaling pathway (Fig. 1e), reflecting that cell-cycle control requires extensive changes of protein as well as phosphoprotein levels. This analysis clearly shows that including phosphorylation profiling in the molecular characterization of cancer cell lines adds a dimension of data that contains information not available when measuring proteomes alone.

**Pathway and kinase activity landscapes of tumor cell lines.** Based on the quantitative (phospho)proteome data, we computed activity landscapes in order to assess the relative activity of signaling pathways (by integrating information on proteins and p-sites involved in a pathway) and kinases (by integrating information on kinase abundance, kinase phosphorylation and kinase-substrate phosphorylation) in cell lines (Fig. 2a, b, Supplementary Fig. 3A, B; Supplementary Methods). Activity landscapes of pathways and kinases (Supplementary Data 5) feature mountains and basins representing high (close to 1) and low (close to 0) relative pathway or kinase activity, respectively. The topography connects cell lines with similar activity profiles and allows quickly identifying the main biology underlying their pathogenic/carcinogenic phenotypes. For example, the mountain WNT signaling in the CRC65 dataset (Fig. 2a) recapitulates that aberrant WNT signaling is frequently driving colorectal cancer[13]. Similarly, hyperactivation of cell-cycle proteins in leukemia cell lines is in-line with their generally short doubling time (Supplementary Fig. 3A). The kinase activity landscapes also recapitulate high ABL1 kinase activity in K562 cells (Fig. 2c) caused by the *BCR-ABL* gene fusion[14] and high ALK activity in SR cells (Fig. 2d) and C10 cells (Supplementary Fig. 3B) that carry the *NPM1-ALK* and *EML4-ALK* gene fusions, respectively[15,16]. More globally, such landscapes efficiently visualize the very large molecular heterogeneity of cancer cell proteomes and phosphoproteomes, which likely reflect their heterogeneous phenotypic characteristics.

**Correlation networks functionalize proteins and p-sites.** While (phospho)proteomic data can be generated efficiently today, the challenge arises that not all proteins and only very few p-sites are functionally annotated. We, therefore, developed a new method incorporating weighted gene correlation network analysis[17] and gene set variation analysis[18] to place proteins and p-sites into new functional and phenotypic contexts simultaneously using a guilt-by-association approach (Supplementary Methods). Figure 2e gives one example for such an analysis for the CRC65 cell lines that are characterized by microsatellite instable (MSI+) or stable (MSI−) genomes. Here, we first concatenated the quantitative protein and p-site data and grouped them by functional annotations such as gene ontology terms or membership in protein complexes into groups of functionally related proteins/p-sites (nodes on 1st axis). Next, we clustered the combined proteins and p-sites by the similarity of their abundance profiles across the CRC65 dataset into groups of abundance-related proteins/p-sites (nodes on 2nd axis). Finally, we grouped cell lines by their MSI status (nodes on 3rd axis). Connections between the three axes represent significant relations between nodes. For example, connections between the 1st and 2nd axis show groups of functionally related proteins/p-sites significantly enriched in groups of abundance-related proteins/p-sites. Similarly, connections between the 1st/2nd and 3rd axis highlight groups of

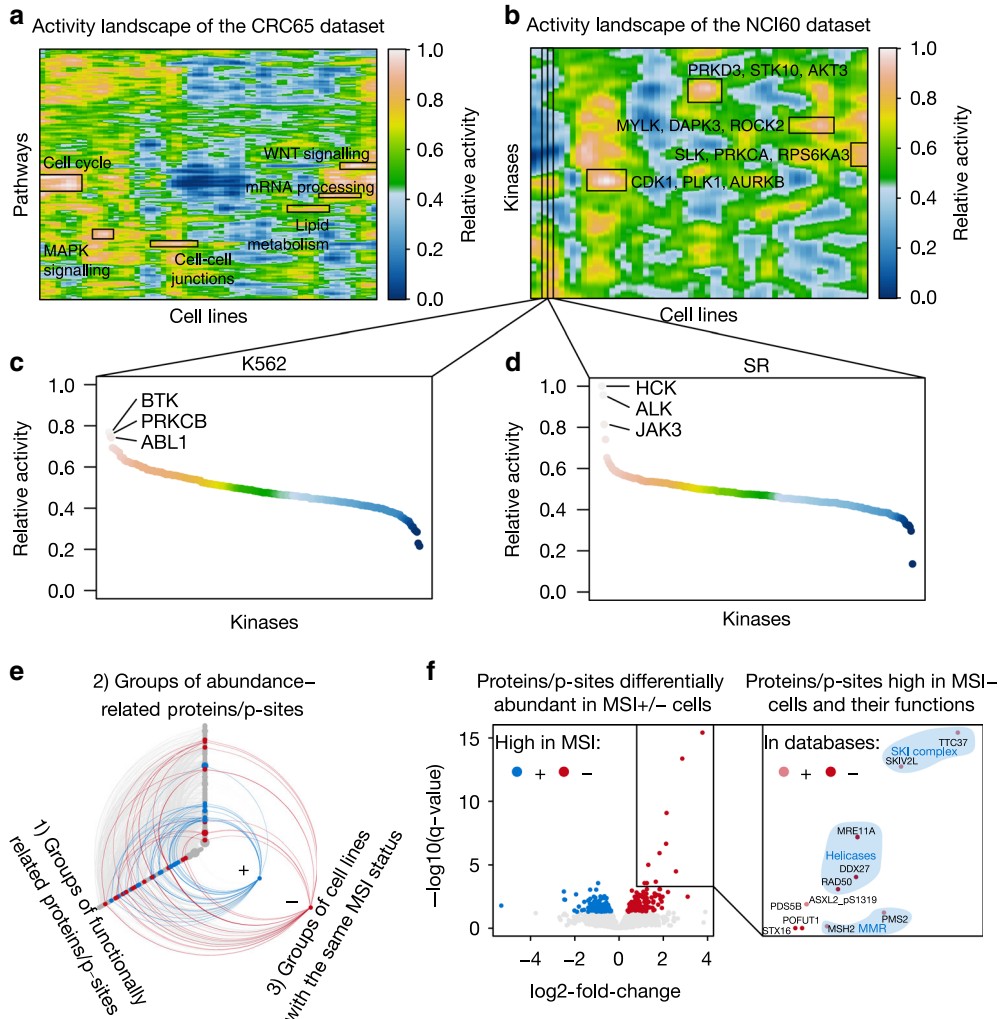

**Fig. 2 Activity landscapes of pathways and kinases functionalize proteins/p-sites. a** Activity landscape of cellular pathways for the CRC65 panel ($n = 64$ cell lines) and **b** of kinases for the NCI60 panel ($n = 60$ cell lines; Supplementary Methods). Relative activity ranges from 0 to 1 representing minimal and maximal relative activity, respectively. Areas of high activity are labelled. **c, d** Waterfall plots visualizing the relative activity of kinases in **c** K562 cells and **d** SR cells. The kinases with the highest relative activity are highlighted. The color of the data points corresponds to the color scale of panel **b**. Relative activity ranges from 0 to 1 representing minimal and maximal relative activity, respectively. **e** Hive plot showing significant associations between groups of functionally related proteins/p-sites, cell lines with the same MSI status and groups of abundance-related proteins/p-sites (highly correlated proteins and p-sites) identified by weighted gene correlation network analysis in the CRC65 dataset (Supplementary Methods). Colored edges (blue = MSI+, red = MSI−) represent significant associations between corresponding nodes on all three axes. **f** Volcano plot visualizing single proteins/p-sites from colored groups of abundance-related proteins/p-sites in **c**, which show significantly higher abundance in MSI+ ($n = 17$; blue) or MSI− ($n = 47$; red) cells (Benjamini–Hochberg corrected $P < 0.05$, moderated $t$-test). Proteins/p-sites in the top right corner are highly abundant in MSI− cell lines and their functional annotations are shown in the magnification (right panel). The light red color in the right panel (In databases+) indicates proteins/p-sites, which are part of the functional annotations enriched in the corresponding group of abundance-related proteins/p-sites from C (e.g., MSH2), while the solid red color (In databases−) highlights proteins/p-sites for which we suggest new functional annotations using guilt-by-association (e.g., ASXL2_pS1319). Shaded areas highlight proteins/p-sites with common functions (blue text). Source data are provided as a Source Data file.

functionally/abundance-related proteins/p-sites with significant differential abundance between groups of MSI+ and MSI− cells, respectively. Colored nodes and connections are full circles. Supporting the significance of this approach, we observed that many proteins/p-sites with significantly higher abundance (FDR < 0.05; fold-changes between 1.3 and 13.7; moderated $t$-statistic) in MSI− cells (red nodes and edges) compared to MSI+ cells are involved in DNA mismatch repair (MMR), while proteins/p-sites with significantly higher abundance (FDR < 0.05; fold-changes between 1.3 and 40.1; moderated $t$-statistic) in MSI+ cells (blue nodes and edges) compared to MSI− cells are primarily involved in transcriptional processes. Focusing on proteins/p-sites with significantly higher abundance in MSI− cells compared to MSI+

cells highlights the DNA helicases MRE11A and RAD50 (involved in double-strand break repair) and the putative RNA helicase DDX27 as associated with MMR proficiency alongside the known MMR proteins MSH2 and PMS2[19] (Fig. 2f). MRE11A and RAD50 are known to be frequently mutated in MSI+ cells[20] and DDX27 was recently described to promote colorectal cancer growth and metastasis[21]. Similarly, a p-site (ASXL2_pS1319) on a protein previously shown to be part of the PR-DUB complex[22] promoting the deubiquitination of histone H2A at lysine 119 and regulating DNA double-strand break repair is significantly higher abundant (FDR = $1 \times 10^{-5}$; fold-change of 2.5; moderated $t$-statistic) in MSI− cells compared to MSI+ cells. These data, together with the fact that proteins involved in DNA double-strand

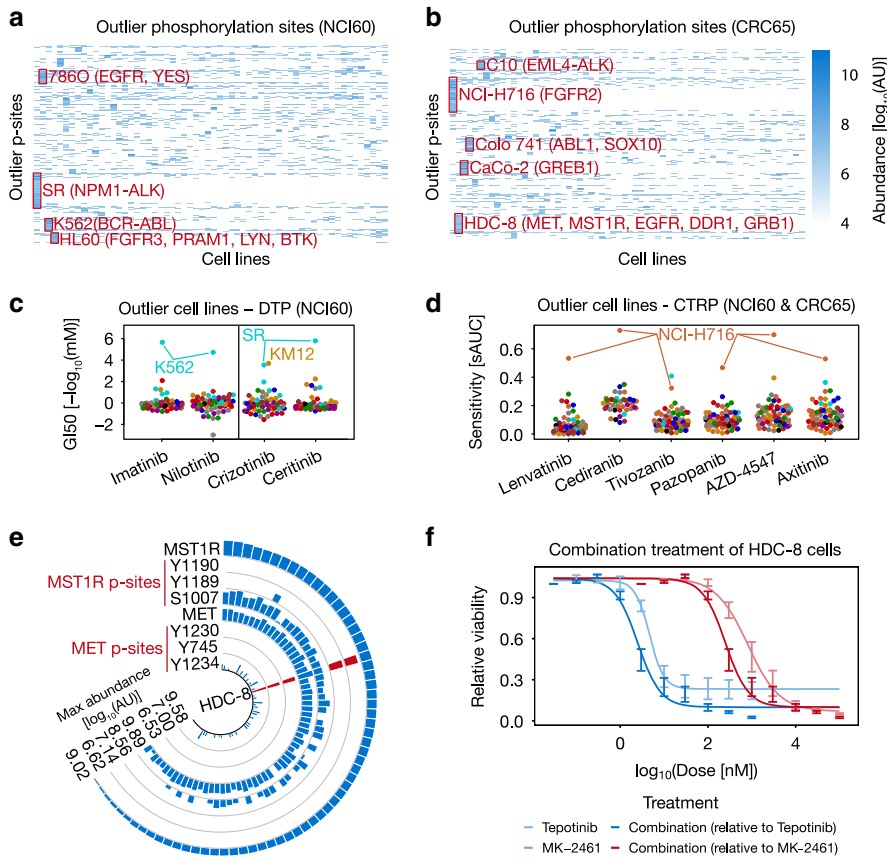

**Fig. 3 Outlier proteins and p-sites abundance often explain drug sensitivity. a, b** Heatmaps of outlier p-sites in the **a** NCI60 ($n = 60$ cell lines) and **b** CRC65 panel ($n = 64$ cell lines). Selected cell lines and p-site clusters are annotated. **c, d** Beeswarm plots visualizing selected drugs for which outlier cell lines were identified in the **c** DTP ($n = 59$ cell lines) and **d** CTRP drug-sensitivity datasets (Lenvatinib: $n = 71$ cell lines; Cediranib: $n = 41$ cell lines; Tivozanib: $n = 72$ cell lines; Pazopanib: $n = 71$ cell lines; AZD4547: $n = 66$ cell lines; Axitinib: $n = 69$ cell lines; sAUC = 1-standardized area under the dose-response curve; GI50 = growth inhibitory concentration analogous to an IC50). Cell lines are colored by tissue of origin as in Fig. 1. **e** Circular bar plot highlighting the outlier abundance of selected pY-sites on MET and MST1R in HDC-8 cells (red bars; $n = 65$ cell lines). **f** Dose-response curves visualizing the synergistic effect of targeting MET (using Tepotinib) and MST1R (using MK-2461) simultaneously in HDC-8 cells. Error bars represent the minimum and maximum relative viability of technical triplicates. Source data are provided as a Source Data file.

break repair are frequently mutated in MSI+ colorectal cancers[23] suggest that ASXL2 may also be mutated in MSI+ colorectal cancers. A similar analysis was performed for the NCI60 cell lines (Supplementary Fig. 3C), and this concept can be readily extended to other phenotypic categories such as the mutational status of known oncogenes. The thousands of functional associations obtained by this analysis can be explored using ATLANTiC to provide testable hypothesis in more specialized experiments (see also Supplementary Data 6).

**Outlier proteins and p-sites often explain drug sensitivity.** Next, we explored the possibility of using protein and p-site abundance to explain drug sensitivity and first focused on outliers (i.e., proteins/p-sites with substantially higher abundance in a particular cell line relative to all other cell lines; Supplementary Methods). Outliers are particularly attractive because of their potential ease of interpretation. Clustering outlier p-site abundance across cell lines reveals hundreds of strong phosphorylation events from the same pathway context in particular cell lines (Fig. 3a, b; Supplementary Data 7). Examples include p-sites on BCR-ABL fusion protein in K562 cell lines (e.g., ABL1_pY204, BCR_pY177, Supplementary Data 7), as well as on pathway members such as LYN_pY193, INPPL1_ pY1162 or STAT5-A_pY90[24]. This provides an additional way of functionalizing p-sites, because p-sites with similar outlier behavior might represent

novel pathway members of hyperactive kinases in the respective cell lines.

Furthermore, this analysis can prioritize potential activity-driven drug response markers. Supporting this notion, the analysis based on p-sites recapitulates that K562 cells harbor high levels of BCR-ABL activity (at least 7-fold higher compared to other cell lines), which explains their exquisite sensitivity towards the BCR-ABL inhibitors Imatinib and Nilotinib (Fig. 3c). Similarly, we observed that KM12 and NCI-H716 cells show high abundance of NTRK1 (e.g., NTRK1_pY398; only detected in KM12 cells) and FGFR2 phosphorylation (e.g. FGFR2_pY378; 10-fold higher compared to other cell lines) and these cells are sensitive to the ALK/NTRK1 inhibitor Crizotinib and designated FGFR/VEGFR inhibitors such as Lenvatinib (Fig. 3c, d), respectively, which was previously only observed at the mRNA level[16]. Another example is the sensitivity of SR cells to Crizotinib and Ceritinib due to high abundance of ALK phosphorylation (e.g., ALK_pY1092; 8-fold higher compared to other cell lines), likely caused by constitutive kinase activity of the NPM1-ALK fusion protein[15] (Fig. 3c). This outlier characteristic is not always as apparent at the mRNA[25] or protein level (Supplementary Fig. 4A), underscoring the value of measuring the phosphoproteome in addition to the proteome and transcriptome of tumor cell lines. In addition, outlier p-site abundance (but not protein abundance) of MET_pY1234, as well as pY1189 and pY1190 on

MST1R isoform 2 suggests that HDC-8 cells may be driven by MET and MST1R activity (Fig. 3e). Given that these p-sites also indicate crosstalk between MET and MST1R[26], we hypothesized that HDC-8 cells may be vulnerable to combined inhibition of these two receptor tyrosine kinases. This is confirmed by observing mild synergy when combining Tepotinib (a selective MET inhibitor) and MK-2461 (an MST1R inhibitor) in in vitro experiments (Combination Index 0.85 at ED$_{50}$; Fig. 3f; Supplementary Fig. 4B). These results show that outlier p-sites can represent (pharmacodynamic) markers for drug sensitivity in cells and can also be useful to select rational drug treatment combinations based on functionally relevant molecular associations.

Phosphorylation aside, there are many examples for protein expression outliers including the drug exporter ABCB1, which is associated with multidrug-resistance in NCIADRRES cells (Supplementary Fig. 4C). Similarly, SLC16A10, which mediates the uptake of aromatic amino acids[27] is only detected in MOLT4 cells (GluC data), which are sensitive to the DNA alkylating agent Bendamustine. Given that Bendamustine is an aromatic acid, we hypothesize that SLC16A10 may be responsible for the sensitivity of MOLT4 cells to this drug (Supplementary Fig. 4D).

**Correlation-based markers of drug sensitivity.** Since prominent outliers at both drug and protein/p-site level are rare in relative terms, we next associated proteins/p-sites with phenotypic drug-sensitivity data using elastic net and random forest regression, as well as by calculating the correlation for protein- or p-site-drug combinations with at least seven pairwise-complete observations (Supplementary Fig. 4E; Supplementary Methods). These analyses cover 872 drugs and resulted in 4,558,128 protein-drug and 6,970,493 p-site-drug associations, which can be explored using ATLANTiC.

At the protein level, elastic net regression reveals that the protein SLFN11 is the most common sensitivity marker in the Developmental Therapeutics Program (DTP) drug dataset (Fig. 4a). Its protein abundance (intensity [AU] fold-change of 172 from the lowest $= 1 \times 10^{6.40}$ to the highest $= 1 \times 10^{8.63}$ value) is highly correlated with topoisomerase inhibitors including Irinotecan ($R = 0.60$, $P$-value $= 1 \times 10^{-6}$), confirming prior observations at the mRNA level[28], as well as with response to DNA synthesis inhibitors such as Gemcitabine ($R = 0.64$, $P$-value $= 2 \times 10^{-7}$) and Triethylenemelamine ($R = 0.62$, $P$-value $= 4 \times 10^{-7}$; Supplementary Fig. 5A, B). This may be rationalized by the recent discovery that SLFN11 blocks stressed replication forks independently of ATR[29] and hints at a potentially fundamental biological role of this poorly studied protein. Focusing on preclinical and clinical drugs allowed the identification of proteomic markers for specific modes of action. We found that the sensitivity to the EGFR inhibitor Cetuximab is associated with low EPHA2 protein abundance (intensity [AU] fold-change of 342 from the lowest $= 1 \times 10^{6.92}$ to the highest $= 1 \times 10^{9.46}$ value; Supplementary Fig. 5C, D), corroborating earlier observations[30]. Another example is ADK, which we found to be a sensitivity marker for Triciribine-5′-monophosphate (intensity [AU] fold-change of 21 from the lowest $= 1 \times 10^{8.81}$ to the highest $= 1 \times 10^{10.12}$ value; Supplementary Fig. 5E, F). The drug is prone to inactivation by extracellular dephosphorylation[31] but can be converted back to its active form by intracellular ADK[32], possibly explaining the increased sensitivity of high ADK expressing cells to this drug. A third example is Arsenic trioxide, which induces oxidative stress in human cell lines[32] (Fig. 4b). Elastic net and network analysis (Supplementary Methods) suggest that high levels of proteins involved in glutathione homeostasis are resistance markers for this compound. For example, we observed that glutathione reductase

(GSR), the catalytic subunit of glutamate-cysteine ligase (GCLC; the first rate-limiting enzyme of glutathione synthesis) and several other members of their molecular network (including the oxidoreductase NQO1) show significant associations with Arsenic trioxide resistance. This suggests that high activity of enzymes involved in glutathione homeostasis can overcome the oxidative stress induced by Arsenic trioxide.

Correlating the p-site data to drug sensitivity revealed that the response to kinase inhibitors can largely be explained by the phosphorylation status of their primary targets or downstream pathway members. For instance, cell lines sensitive to Selumetinib (a MEK inhibitor) show high phosphorylation of RPS6KA3_pS715 (RSK2; intensity [AU] fold-change of 23 from the lowest $= 1 \times 10^{7.04}$ to the highest $= 1 \times 10^{8.39}$ value), which is downstream of MEK-MAPK signaling (Fig. 4c). In contrast, high phosphorylation of T581 on the GATOR complex protein WDR24 (intensity [AU] fold-change of 12 from the lowest $= 1 \times 10^{6.52}$ to the highest $= 1 \times 10^{7.59}$ value) is a resistance marker for Selumetinib. We hypothesize that WDR24_pT581 may inhibit the GATOR1 sub-complex, which suppresses the responsiveness to MEK inhibition via activating mTORC1 and the TORC1 signaling pathway[33,34]. Similarly, MAPK3_pY204, MAPK1_pY187, as well as RPS6KA1_pT368 are associated with resistance to Perifosine (an AKT inhibitor; intensity [AU] fold-change of at least 30 from the lowest $= 1 \times 10^{6.40}$ to the highest $= 1 \times 10^{7.88}$ value; Fig. 4d, e). The former two sites are well-known indicators of kinase activity[35]. Recently, it was shown that phosphorylation of MAPK1/3 decreases upon AKT-induced phosphorylation of FOXO1[36]. Hence, high abundance of MAPK3_pY204 and MAPK1_pY187 might be indicators of low AKT activity and thus resistance to AKT inhibition. In addition, BRAF_pS151 is associated with resistance to the IGF1R inhibitors Linsitinib and BMS-754807 (intensity [AU] fold-change of 151 from the lowest $= 1 \times 10^{6.64}$ to the highest $= 1 \times 10^{8.82}$ value; Supplementary Fig. 5G, H). Earlier work has shown that Insulin receptor upregulation is a resistance mechanism for BRAF inhibitors[37], suggesting that when BRAF_S151 is not phosphorylated, cells may be sensitive towards IGF1R inhibition and conversely, when BRAF_S151 is phosphorylated, co-inhibition of the two kinases may have favorable effects.

To study the potential of combining kinase inhibitors with other drugs systematically in silico, we correlated the abundance of activity-enhancing or inhibitory p-sites on kinases with the response to all drugs (Supplementary Fig. 5I). This revealed that e.g. CHEK1_pS186 (an inhibitory site) is positively correlated with Artesunate sensitivity, suggesting that co-treatment with Artesunate and CHEK1 inhibitors may have a synergistic effect. This agrees with the observation that Artesunate generates oxidative DNA damage, which triggers the activation of CHEK1 and other DNA damage proteins[38]. We observed that most of the p-site-drug associations are not detectable at the proteome level. Therefore, the extensive phosphorylation data presented here is a unique resource complementary to other omics data, which can be used to investigate complex tumor biology and study the mode of action (MoA) of drugs.

Next, we asked the question whether the sensitivity of cell lines to drugs can be explained by the levels of proteins that are involved in their metabolism. As an example, we trained random forest models (Supplementary Methods) on the CRC65 dataset using all combinations of 14 proteins involved in the metabolism of the antimetabolite drug 5-fluorouracil (5FU; frequently used to treat colorectal cancer) in order to capture non-linear relationships between protein abundance and drug sensitivity. Models using 3–7 proteins achieve good prediction accuracy in colorectal cancer cell lines of the NCI60 panel ($R \sim 0.7$; Fig. 4f,

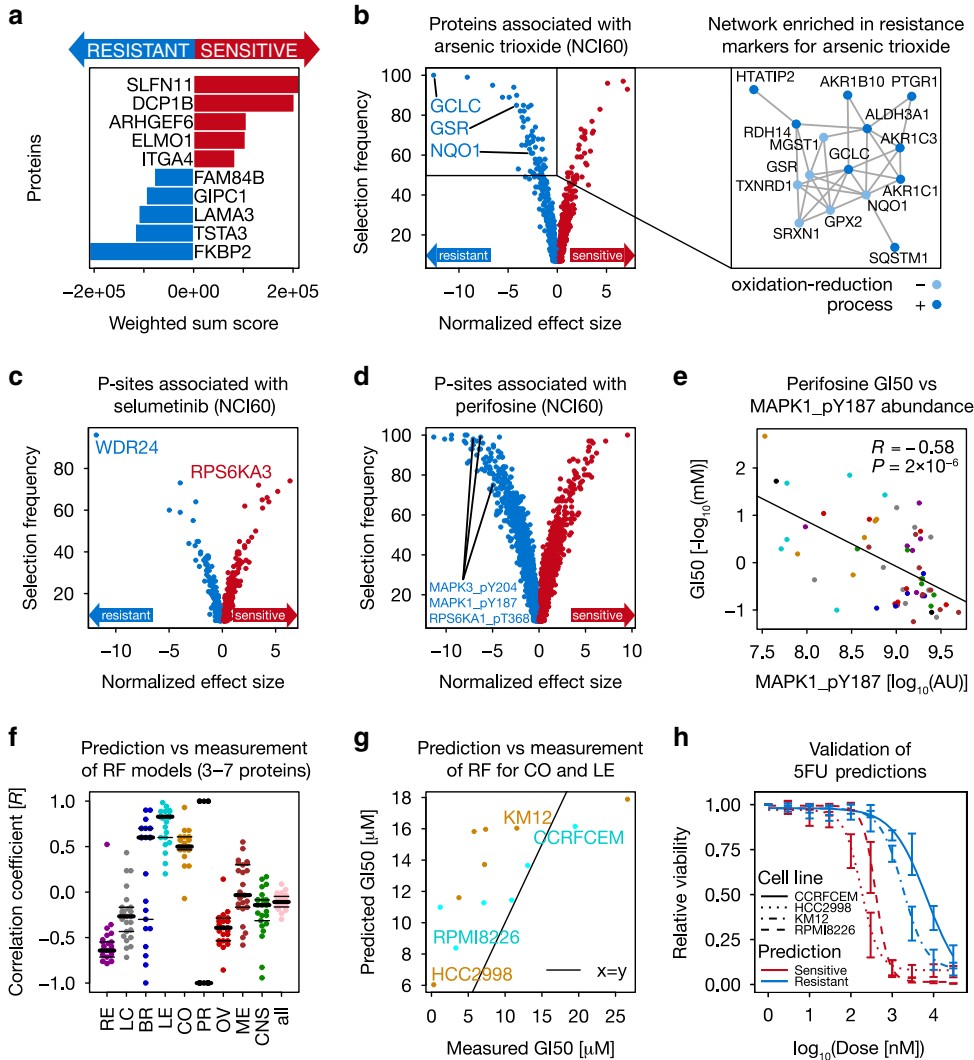

**Fig. 4 Correlation-based (phospho)proteome markers often explain drug sensitivity. a** General sensitivity and resistance protein markers of the NCI60 panel in the DTP drug response dataset (Supplementary Methods). **b** Volcano plot of protein markers associated with response to Arsenic trioxide in the NCI60 panel (Supplementary Methods). GCLC, GSR, and NQO1 are resistance markers (left panel). Right panel: highly connected network (according to https://string-db.org/) enriched by Arsenic trioxide resistance markers. **c**, **d** Volcano plots of p-sites associated with **c** Selumetinib and **d** Perifosine sensitivity and resistance (Supplementary Methods). **e** Significant negative correlation between Perifosine sensitivity and MAPK1_pY187 abundance (*n* = 60 cell lines; Pearson correlation; *P* < 0.05). Cell lines are colored by tissue of origin as in Fig. 1. **f** Prediction accuracy (Pearson correlation) between predicted and measured 5FU sensitivity of the top 5 random forest models combining 3–7 proteins involved in 5FU metabolism (*n* = 25 models each; Supplementary Methods). **g** Correlation between predicted (using a random forest model) and measured 5FU sensitivity for leukemia and colorectal cancer cell lines (*n* = 13 cell lines; DTP data; not used for training). KM12 (colon) and CCRFCEM (leukemia) are resistant cell lines, while HCC2998 (colon) and RPMI826 (leukemia) are sensitive cell lines (GI50 = growth inhibitory concentration analogous to an IC50). Cell lines are colored by tissue of origin as in Fig. 1. **h** Cell viability assay confirming the predicted drug sensitivity (red) and resistance (blue) of cell lines from panel **g**. Error bars represent the minimum and maximum relative viability of technical triplicates. Source data are provided as a Source Data file.

Supplementary Data 7), but not in most other entities. Interestingly, the models trained on CRC cell lines perform even better in predicting sensitivity to 5FU for leukemia cell lines. Therefore, we performed additional in vitro experiments based on the two best 5-protein models (which shared TYMP, RRM1, UPP1, and UCK1 and contained either PPAT or UCK2) using two CRC and two leukemia cell lines that are predicted to be sensitive or resistant to 5FU (Fig. 4g). In-line with the prediction, the leukemia cell line RPMI8226 ($ED_{50}$ 0.40 μM) was 17 times more sensitive to 5FU than CCRFCEM cells ($ED_{50}$ 6.96 μM; Fig. 4h). The same held true for the selected colorectal cancer cell line pair ($ED_{50}$ 0.18 μM in HCC2998 vs. $ED_{50}$ 1.83 μM in KM12), suggesting that modeling drug response as a function of proteins

involved in a drugs' metabolism can be a powerful tool for predicting drug sensitivity.

**Identifying markers for drugs with the same mode of action**. So far, we analyzed each drug in isolation in search of (phospho) proteomic markers for drug response. Based on the hypothesis that the responses to drugs targeting the same pathways might be explained by the same group of molecular phosphoprotein/p-site markers, we applied an extended sparse multiblock partial least square regression (SMBPLSR) algorithm[39], which clusters drugs sharing similar phenotypic profiles and associates (the intensities of) multiple phosphoproteins and/or p-sites that show divergent abundance patterns with these drugs (Supplementary Methods).

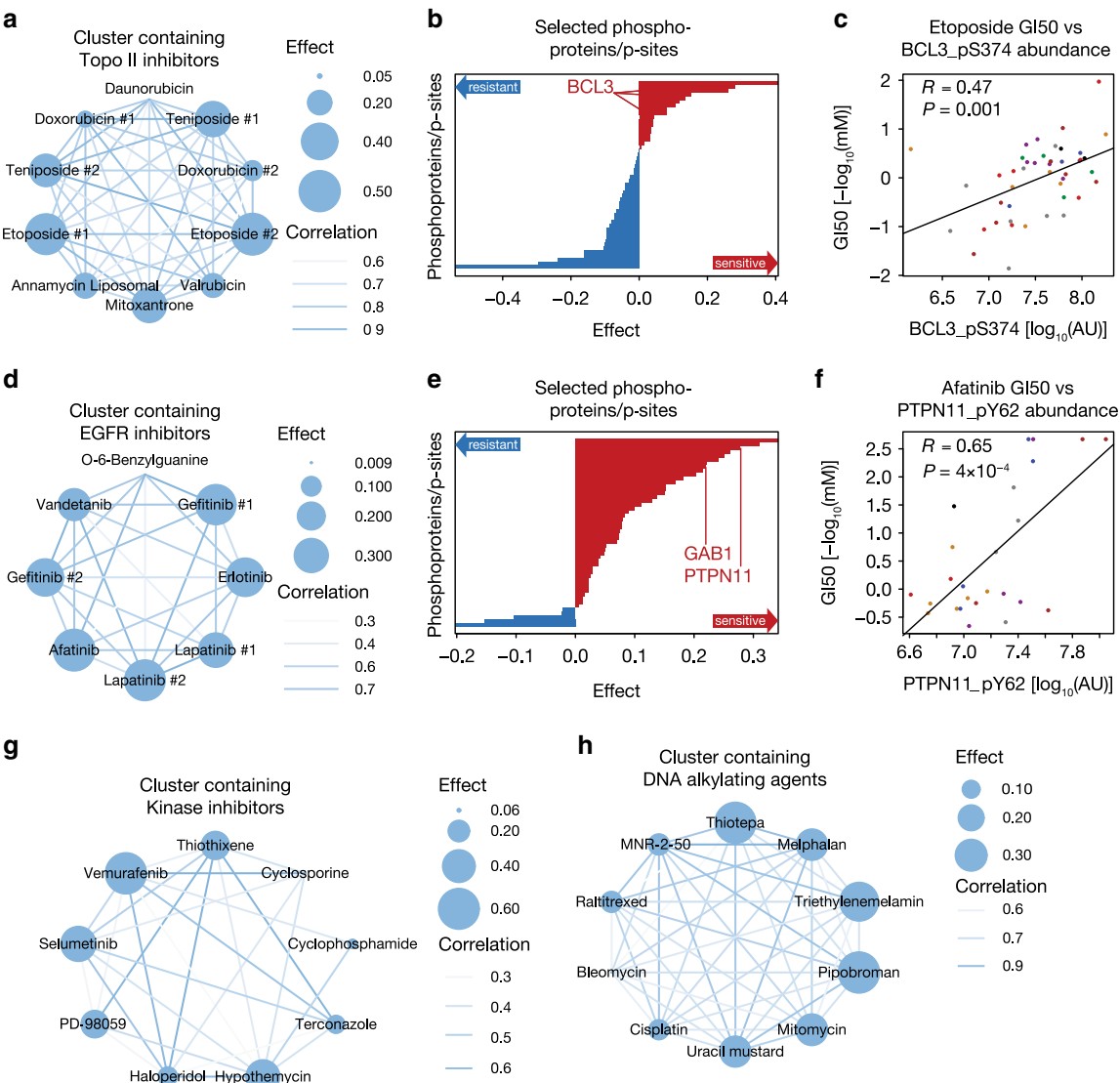

**Fig. 5 Common markers explain sensitivity to drugs sharing the same mode of action. a** Cluster containing several Topoisomerase II inhibitors identified by sparse multiblock partial least square regression (SMBPLSR). Nodes represent compounds with size indicating effect size (SMBPLSR score of drugs; Supplementary Methods), while edges visualize the correlation of the sensitivity between two drugs across cell lines (white = low, blue = high). **b** The bar plot shows phosphoprotein and p-site markers associated with drugs in the network depicted in the left panel (red = sensitivity marker, blue = resistance marker). **c** Selected markers include multiple p-sites on BCL3, one of which (BCL3_pS374) is significantly (Pearson correlation test; $P < 0.05$) positively correlated with Etoposide sensitivity (right panel; $n = 52$ cell lines). Cell lines are colored by tissue of origin. **d** Same as in panel **a**, but for the drug cluster containing mainly EGFR inhibitors. **e** Here, GAB1 and PTPN11 phosphoproteins are selected as sensitivity markers (middle panel). **f** PTPN11_pY62 shows significant (Pearson correlation test; $P < 0.05$) positive correlation with Afatinib (right; $n = 52$ cell lines). (**g**, **h**) Networks of drug clusters driven by **g** kinase inhibitors **h** DNA alkylating agents. Source data are provided as a Source Data file.

This way, we identified for example high relative abundance of multiple p-sites on BCL3 (intensity [AU] fold-change of at least 140 from the lowest $= 1 \times 10^{5.90}$ to the highest $= 1 \times 10^{8.05}$ value), a regulator of cell proliferation and apoptosis[40], as sensitivity markers for topoisomerase II inhibitors, such as Teniposide, Etoposide, and Doxorubicin (Fig. 5a–c). Similarly, the sensitivity of multiple EGFR inhibitors (e.g., Lapatinib, Gefitinib, and Afatinib; Fig. 5d–f) is associated with high relative abundance of p-sites in the EGFR signaling pathway. These include GAB1_pY689 (intensity [AU] fold-change of 9 from the lowest $= 1 \times 10^{6.61}$ to the highest $= 1 \times 10^{7.55}$ value), the primary mediator of EGF-stimulated activation of PI3K/AKT[41], and PTPN11_pY62 (intensity [AU] fold-change of 27 from the lowest $= 1 \times 10^{6.61}$ to the highest $= 1 \times 10^{8.05}$ value), a phosphatase that dephosphorylates GAB1 and EGFR[42]. Several kinase inhibitors form a

strong cluster that is enriched in drugs targeting the BRAF-MEK-ERK1/2 axis, including Vemurafenib (BRAF), Selumetinib (MEK), PD98059 (MEK), and Hypothemycin (MAPK1/ERK3; Fig. 5g). The associated sensitivity markers include STK4_pS410 and multiple p-sites on the transcription factor SOX10 (including T240). A recent study has shown that ERK1/2 suppress the transcriptional activity of SOX10 through phosphorylation of T240 in BRAF mutant cell lines. Consequently, transcriptional targets of SOX10, including FOXD3, which mediates adaptive resistance to RAF inhibitors, were downregulated[43]. Our analysis suggests that high abundance of SOX10_pT240 (intensity [AU] fold-change of 446 from the lowest $= 1 \times 10^{6.27}$ to the highest $= 1 \times 10^{8.92}$ value) is associated with sensitivity to BRAF, MEK, and ERK inhibitors. These results underscore the importance of understanding the signaling network of kinases in detail, since

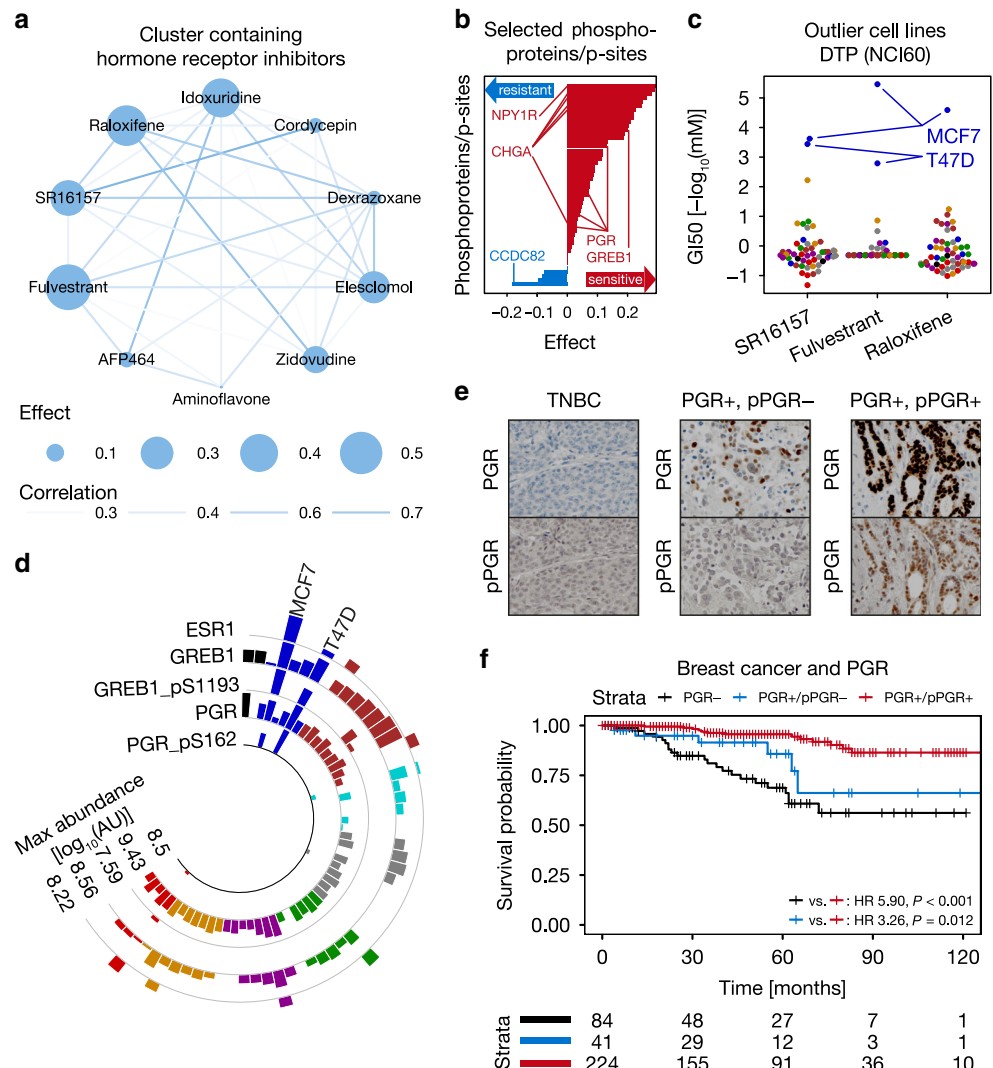

**Fig. 6 High phosphorylation of PGR predicts benefit from endocrine therapy in breast cancer. a** Cluster containing several hormone-receptor inhibitors identified by sparse multiblock partial least square regression (SMBPLSR; $n = 10$ drugs). Nodes represent compounds with size indicating effect (SMBPLSR score of drugs; Supplementary Methods), while edges visualize the correlation of the sensitivity between two drugs across cell lines (white = low, blue = high). **b** Bar plot showing examples for phosphoprotein or p-site markers associated with drugs in panel **a** including PGR (pS162) and GREB1 (pS1193; sensitivity markers in red, resistance markers in blue). **c** Sensitivity of NCI60 cell lines towards hormone-receptor inhibitors (SR16157: $n = 52$ cell lines; Fulvestrant: $n = 52$ cell lines; Raloxifene: $n = 53$ cell lines). MCF7 and T47D cells are particularly sensitive outlier cell lines. Cell lines are colored by tissue of origin as in Fig. 1. **d** Circular bar plot showing relative protein expression of ESR1, GREB1, and PGR, as well as the abundance of GREB1_pS1193 and PGR_pS162 across the NCI60 panel ($n = 57$ cell lines). Cell lines are colored by tissue of origin as in Fig. 1. **e** Representative pictures of breast cancer tissue microarrays (TMAs) stained for PGR protein and PGR_pS162 (pPGR). Triple-negative breast cancer (TNBC) cases served as negative controls. **f** Kaplan–Meier plot of breast cancer patients showing that PGR+/pPGR+ patients survive significantly longer than PGR+/pPGR− and PGR- patients (log-rank test). The number of patients at risk is shown below the Kaplan–Meier plot. Source data are provided as a Source Data file.

inhibiting an upstream kinase might result in pathway rewiring, which could be avoided by targeting the downstream protein on which these different pathways converge. Additional clusters are driven by other MoAs (e.g., DNA alkylating agents and anti-metabolites; Fig. 5h) or contain drugs with different or unknown MoAs. Markers identified by SMBPLSR, which cannot be directly linked to the MoA of the drugs with which they are associated thus help us to improve our understanding of why different drugs may have similar sensitivity profiles across the cell lines (Supplementary Data 8).

**Response of phospho-PGR+ patients to endocrine cancer drugs.** In order to validate potential biomarkers suggested by SMBPLSR, we focused on a cluster of drugs containing several

modulators of estrogen signaling (e.g., Raloxifene; Fig. 6a), which are primarily used for the treatment of hormone-receptor-positive breast cancer[44]. Progesterone receptor (PGR) phospho-protein abundance as well as several p-sites (pS20, pS81, and pS162) are associated with this cluster of drugs, as well as GREB1_pS1193 (Fig. 6b). Even though the function of this particular p-site is not known, GREB1 has been found to be important for ESR1 signaling[45]. The breast cancer cell lines MCF7 and T47D are very sensitive to a selection of antihormonal drugs (Fig. 6c), which prompted us to investigate whether ESR1, PGR, and GREB1 protein show similar abundance profiles across the NCI60 panel as their p-sites. Interestingly, protein abundance fails to set MCF7 and T47D apart from the remainder of the panel, while both GREB1_pS1193 and PGR_pS162 clearly establish them as outlier cell lines (Fig. 6d). It is known that the

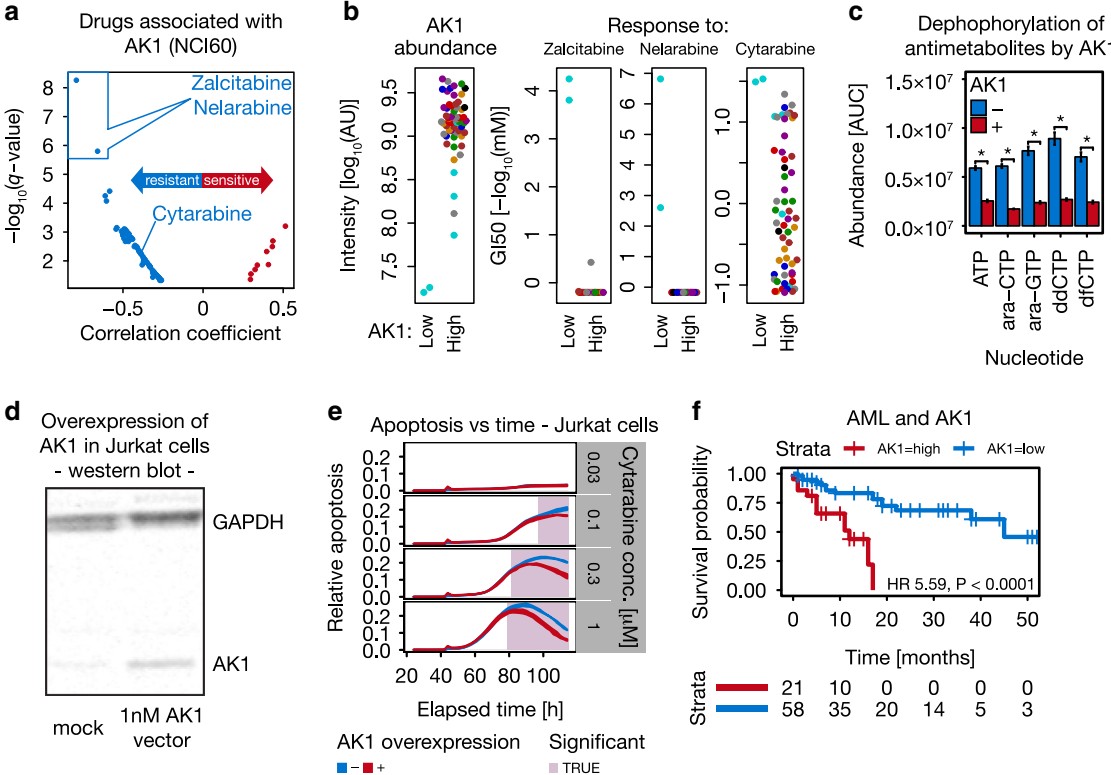

**Fig. 7 High AK1 levels signify chemotherapy resistance in AML. a** Correlation analysis (Supplementary Methods) identified AK1 protein expression to be strongly negatively correlated with response to antimetabolite drugs including Zalcitabine and Nelarabine. **b** AK1 expression (left panel) and drug-sensitivity analysis of Zalcitabine, Nelarabine and Cytarabine in the NCI60 panel (right three panels; $n = 59$ cell lines). Cell lines are colored by tissue of origin as in Fig. 1. **c** Quantification of antimetabolites by multiple reaction monitoring (Supplementary Methods) in their active nucleotide triphosphate form (NTP) showing that AK1 decreases NTP abundance significantly in vitro (t-test; $P < 0.05$; ara-CTP = Cytarabine; ara-GTP = Nelarabine; ddCTP = Zalcitabine; dfCTP = Gemcitabine). ATP served as a positive control. Error bars represent the standard deviation of three biological replicates. **d** Western blot confirming successful overexpression of AK1 in Jurkat cells (Supplementary Methods). **e** AK1 overexpression (red) in combination with Cytarabine treatment in Jurkat cells resulted in significantly reduced apoptosis in a concentration- and time-dependent manner (time-point-wise t-test, $P < 0.05$). The width of blue and red lines represents the range of relative apoptosis across three technical replicates. **f** Kaplan–Meier analysis of AML patients stratified by AK1 expression. Patients with low AK1 expression (blue) survive significantly longer than patients with high AK1 expression (red; log-rank test, $P < 0.0001$). The number of patients at risk is shown below the Kaplan–Meier plot. Source data are provided as a Source Data file.

phosphorylation of S162 induces the transcriptional activity of PGR[46]. Therefore, we hypothesized that, in addition to the clinically established measurement of hormone-receptor expression as a selector for antihormonal therapy in breast cancer[47], the activity status of PGR could potentially further stratify hormone-receptor-positive patients and may predict sensitivity to anti-hormonal therapy in this preselected patient population. To lend support for this hypothesis, we performed immunohistochemistry staining of PGR and PGR_pS162 in a cohort of 361 breast cancer patients with known hormone-receptor status (PGR +/− and/or ER +/−; Fig. 6e, Supplementary Methods). As expected, hormone-receptor-positive patients (here exemplarily selected by PGR positivity) who received endocrine therapy +/− chemotherapy survive significantly longer than hormone-receptor-negative patients who received alternative therapies (Fig. 6f; Supplementary Methods). Interestingly, individuals with high levels of phosphorylation of PGR (PGR_pS162 H-scores ≥ 110; Supplementary Methods) among PGR+ patients show even longer survival (Fig. 6f), which is, however, only marginally statistically significant (HR = 3.26, P-value = 0.012, log-rank test). Cox proportional-hazards models were used to evaluate potential confounding factors, revealing that pPGR is confounded by age (P-value = 0.06) and the T grading scale (P-value = 0.10) but independent of tumor grade (P-value = 0.01) and HER2 status (P-value = 0.02; Supplementary Data 9). While these initial

findings are interesting, they need to be reproduced in an independent cohort before they can be considered for further translational research.

**High AK1 levels determine chemotherapy resistance in AML.** Encouraged by the potential clinical translatability of the phosphoproteomics data on breast cancer, we also mined the proteomics data for proteins with potential functional implications in the metabolism of antimetabolites commonly used in chemotherapy. One of the proteins frequently expressed at low levels in antimetabolite-sensitive cell lines (e.g., MOLT4 and CCRFCEM with intensity [AU] fold-changes of at least 0.22 and 0.25, i.e., about 4-fold lower compared to other cell lines, respectively) is Adenylate kinase isoenzyme 1 (AK1). AK1 acts as a balancing enzyme for cellular nucleotide ratios by transferring the terminal phosphate of NTPs to AMP or dAMP[48] and also harbors broad nucleoside diphosphate kinase activity[49]. Interestingly, the two drugs that are most strongly negatively correlated with AK1 abundance (Fig. 7a) are Zalcitabine and Nelarabine (Cytarabine showed a lower but clear correlation; R = −0.39 for Cytarabine versus R = −0.79 for Zalcitabine). This observation is driven by two leukemia cell lines (Fig. 7b) and led to the hypothesis that AK1 may dephosphorylate and thereby inactivate this class of drugs as has recently been reported for SAMHD1 and Cytarabine[50]. In vitro dephosphorylation assays

using recombinant AK1 show that AK1 can indeed reduce the levels of several tri-phosphorylated nucleotide analogues (Fig. 7c; Cytarabine/ara-CTP, Nelarabine/ara-GTP, Zalcitabine/ddCTP and Gemcitabine/dfCTP; Supplementary Data 9). In-line with these results, overexpression of *AK1* (Fig. 7d) in Cytarabine-sensitive Jurkat cells renders the cells more resistant (Fig. 7e). To test if this phenomenon may be relevant in clinical practice, we analyzed a cohort of 79 Cytarabine-treated AML patients for AK1 expression using immunohistochemistry (tumor samples were taken at diagnosis before therapy). We found that patients with low AK1 expression (IHC staining scores of 0 and 1; Supplementary Methods) have a very significantly higher 4-year survival probability than patients with high AK1 expression (IHC staining scores of 2 and 3; Fig. 7f). High abundance of AK1 is a significant predictor for shorter survival even when potential confounding factors such as the mutational status of FLT3 or NPM are included in Cox proportional-hazards models (Supplementary Data 9). We performed a similar analysis for AK1 in the TCGA dataset on the transcript level using www.oncolnc.org[51]. When splitting the TCGA AML cohort according to the expression of *AK1* mRNA into high and low expressing groups in the same ratio as our AML cohort, the findings for AK1 are confirmed at the transcript level. These results indicate that AK1 may dephosphorylate nucleoside analogue drugs in vivo, thereby rendering them less effective. As such, AK1 expression would appear to be an attractive patient stratification biomarker that may enable physicians to administer appropriate drug doses. AK1 may also represent a drug target for combination with nucleoside analogue chemotherapy.

## Discussion

Taken together, our study shows that activity landscapes of tumor cell lines as measured by quantitative (phospho)proteomics can make considerable contributions to understanding and predicting the response of cancer cell lines to a wide variety of cancer drugs. In prototypical examples, we show the potential of the data and of this approach for basic science and translational research, albeit stressing again that the findings on the breast cancer and AML patient cohorts need further validation in independent cohorts before they may be translated into clinical practice. We also anticipate that many more such opportunities may be identified when engaging the scientific community via ATLANTiC and ProteomicsDB. This work also highlights the pressing need for the functionalization of the tens of thousands of p-sites that can nowadays be measured relatively easily. While we demonstrate some ways in which this may be done using bioinformatics, further mechanistic insights into how phosphorylation regulates molecular processes in cancer cells and how this explains or influences the action of drugs will require additional efforts in the future. In particular, the glaring lack of systematic information on kinase-substrate relationships needs addressing with great urgency.

**Reporting summary**. Further information on research design is available in the Nature Research Reporting Summary linked to this article.

## Data availability

Annotated spectra were uploaded to ProteomicsDB[10]. Additionally, all raw mass spectrometry files and MaxQuant result files are available via PRIDE and ProteomeXchange[9] using the accession code PXD013615. The source data underlying Fig. 1c–e, 2–7 and Supplementary Figs. 1–5 are provided as a Source Data file. Our previously published full proteome data on the CRC65 cell line panel[7] have the accession code PXD005354. Source data are provided with this paper.

## Code availability

The code supporting the current study is available on github here https://github.com/kusterlab/ATLANTiC and we will help our readers to reproduce the analyses presented in the current study if the need arises. Source data are provided with this paper.

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

## Acknowledgements

We thank the Comparative Experimental Pathology Platform (CEP) of the Institute of Pathology at the Technical University of Munich for the immunohistochemistry staining of the breast cancer cohort, Andrea Hubauer for expert laboratory assistance and Prof. Dr. Stephan Michael Feller for supplying the CRC65 cell line panel. This study was in part supported by grants from the Deutsche Forschungsgemeinschaft (DFG, German Research Foundation)—EXC 114/3 and SFB 1309—and an ERC Advanced Grant (TOPAS; 833710) to B.K.

## Author contributions

Conceptualization, M.F., C.M., B.R., and B.K.; Methodology, M.F., C.M., B.R., T.O., K.K., E.D., and S.H.; Software, M.F., C.M., P.S., and M.W.; Formal analysis, M.F., C.M., and B.R.; Investigation, M.F., B.R., T.O., S.S., K.K., E.D., A.H., D.H., J.M., and J.Z.; Resources, J.D., H.-M.K., H.S., W.W., and B.K.; Data curation, M.F., C.M., and B.R.; Writing—Original Draft, M.F., C.M., B.R., and B.K.; Writing—review and editing, M.F., C.M., and B.K.; Visualization, M.F., C.M., and B.R.; Supervision, H.-M.K., H.S., W.W., and B.K.; Project administration, B.K.; Funding acquisition, B.K.

## Competing interests

M.W. and B.K. are founders and shareholders of OmicScouts GmbH and msAId GmbH. They have no operational role in these companies. M.F. is a founder, shareholder and the CEO of msAId GmbH. The work presented here was not in any way related to these affiliations. The remaining authors declare no competing interests.
