## [Peer Review File · Nature Communications]

This manuscript has been previously reviewed at another journal that is not operating a transparent peer review scheme. This document only contains reviewer comments and rebuttal letters for versions considered at *Nature Communications*.

Detailed response to reviewer comments:

The authors are grateful to the comments made by the reviewers. The new format of the article allowed us to use more space for data analysis, figures and text all of which have made the manuscript much stronger. The authors hope that all concerns have been adequately addressed. Specific details are provided below but the most significant changes to the manuscript are:

- *We revised the text to clearly reflect that very large quantities of new data were acquired specifically for this manuscript.*
- *Where appropriate, we included explanations of the methods we used throughout the main text in order to ensure a clear reading experience.*
- *We performed Cox Proportional Hazards Regression to evaluate potential confounding factors of our pPGR and AK1 findings (Supplementary Table 9).*
- *We combined Figures 5F and G into Figure 6F in order to make our findings with respect to pPGR clearer.*
- *We added two new panels Figure 2C and D in order to better explain the results of our ATLANTiC score and to highlight our findings from Figure 2B.*

Referees' comments:

Referee #1: cancer drug screens (Remarks to the Author):

Frejno, Meng, Ruprecht, and colleagues present a new analysis of comprehensive proteomic and phospho-proteomic datasets of the NCI60 and CRC65 cell line panels that they have previously published.

The original text might have not made it sufficiently clear that the phosphoproteomics data for all 125 cell lines was specifically acquired for this manuscript. In addition, we also acquired new full proteome data for the NCI60 panel for this manuscript. The data from our previous publication on the full proteomes of the NCI60 panel were not used in the current manuscript as the new data is much more comprehensive. The only proteomic dataset which was previously published and reanalysed for this manuscript was the full proteome data of the CRC65 panel. We changed the manuscript to make this clearer.

They show that global protein and phospho-protein abundance can cluster several groupings of cell lines according to their broad cancer lineage of origin, while others are more heterogeneous. In general, they report strong correlation between total and phospho-peptides, however, many of the associations they observe at the phospho-peptide level are not recorded in pathway databases. They reveal that cell lines can be clustered according to high and low activity of various cellular pathways, and many of these correspond to known genetic alterations in signaling pathways. By mapping correlated abundance levels of unknown and known proteins, predicted functions for unknown proteins can be inferred, as is illustrated by comparing cell lines that are MSI+ versus MSI-. Protein and phospho-protein abundance is also correlated with drug sensitivity, and numerous instances of positive correlations are reported. This correlation with single-agent activity is extended to make predictions of drug combination synergy, one of which is tested. Finally, abundance of PGR phospho-peptides and AK1 total peptides are correlated with sensitivity to estrogen receptor inhibitors in breast cancer and cytarabine in AML, respectively. Both of these observations are validated with IHC on patient sample cohorts with known clinical outcomes.

We thank the reviewer for the concise summary of our manuscript.

1. Many of the broad cancer types that cluster homogeneously in Figure 1 are actually derived from diverse subsets of those cancer types. If the broad cancer subtypes are clustered individually, do more discrete subsets emerge as differentially clustered from one another?

This is an interesting question which we feel cannot be answered by our data set, because of the limited number of samples per cancer type (maximum of 9 cell lines for melanoma and lung cancer). Our study was set up to use the NCI60 panel to get a broad overview over many different cancer types in terms of their (phospho-) proteome in order to then focus on colorectal cancer by using the CRC65 panel. For the CRC65 panel, we already performed a similar analysis as suggested by the reviewer, which was published before (Frejno et al., Mol Systems Biol 2017). Acquiring (phospho-) proteomics data on additional cell lines of the cancer types referred to by the reviewer in order to be able to perform 'sub'-clustering would be interesting but beyond the scope of this already extensive manuscript.

2. The synergy shown in figure 3F is relatively weak, and the data for a full matrix of drug combinations at various ratios is not shown.

The authors acknowledge that the synergy is relatively weak. To clarify, we did indeed not test the full matrix of drug combinations. Instead, we followed the Chou-Talalay method for drug combination studies, which is based on the median-effect equation (Chou, Cancer Research 2010). As recommended before (Chou, Leuk Lymphoma 2008), we chose several concentrations above and below the IC_{50} of the two drugs that were tested for synergy. Combining the two drugs at their IC_{50} 's ratio ensures that the contribution of the two compounds to the effect in the combination is about equal.

It would be worth exploring this drug combination more extensively and/or testing one of a number of other possible synergistic combinations that are suggested in the narrative (e.g. artesunate/CHEK1, etc.). Generally, it would be nice to see more thorough validation of the predictions that are made in the narrative.

The current manuscript primarily presents a resource of proteomic data for many cancer cell lines (clearly the largest to date) and how it may be used. Therefore, instead of working out a few use case more extensively, we decided to present more use cases but with fewer validated examples in order to appeal to a broad readership (e.g. AK1 and pPGR). Therefore, while clearly very interesting, we feel that validating further associations between phosphosites or proteins and drug sensitivity would be beyond the scope of this manuscript. Throughout the manuscript, we state several times that we make our data available and accessible (via atlantic.proteomics.wzw.tum.de) to the scientific community in order to encourage data mining and follow-up work by other laboratories of the type suggested. Again, our manuscript aimed to highlight what may be discovered using our data on a few selected and validated examples.

3. The list of hormone-receptor associated drugs should be double-checked. At least one on the figure, elesclamol, is not associated with targeting of hormone receptors (to the best knowledge of this reviewer).

The authors apologize for the potentially misleading wording. We changed the manuscript to reflect the fact that the cluster merely contains several modulators of estrogen signalling.

4. The improved prognostic value of pPGR is not clear in Figure 5. The outcome of pPGR+ cases appears very similar to PGR+ cases. The cohort should be sub-divided into pPGR+/PGR+, pPGR-

/PGR+, and pPGR-/PGR- to illustrate the improved prognostic potential of the phosphorylated marker.

We agree that the prognostic value of pPGR was not clear from the initial Figure, which is why we reworked Figure 5F and G and combined the two panels into a single panel in Figure 6F. We think that it is now clear that patients in this particular breast cancer cohort, who were PGR+ but did not show PGR phosphorylation (pPGR-) have a lower chance of survival than patients who were PGR+ and pPGR+.

5. The AK1 expression correlation with AML outcome is interesting. It would be worthwhile to see whether the expression correlates with prognostic risk categories and/or with patient subsets defined by various genetic lesions that are known to confer better or worse outcomes.

The authors agree that the AK1 finding is indeed very interesting as this may become clinically actionable. In order to test whether the expression of AK1 is confounded by other prognostic risk factors or genetic lesions, we performed Cox Proportional Hazards Regression (COXPH) analyses on the AML cohort. We modelled survival probability as a function of AK1 in combination with various potentially confounding factors such as Age, Sex and risk factors such as FLT3-ITD. The results of this analysis are now part of Supplementary Table 9, which highlights that, together with low AK1 expression, only the level 'complete remission (CR)' of the factor 'Status CR after induction therapy' and the level 'received stem cell transplantation' of the factor 'Stem cell transplantation' had a significant Hazard ratio below one, indicating a favourable effect on the survival probability.

Also, can AK1 expression at the transcript level be correlated with AML outcome from any of the publicly available AML datasets or is this observation only seen at the protein level?

We performed this analysis using <http://www.oncolnc.org> (Anaya J, PeerJ Computer Science 2016). If we split the LAML cohort according to the expression of AK1 transcript into 'high' and 'low' expressing groups in the same ratio as our AML cohort, then we can reproduce our findings for AK1 at the transcript level. While this is good confirmatory evidence, the authors think that directly measuring protein levels is advantageous as it takes out the many uncertainties in extrapolating protein levels from transcript levels. We included a note on this in the manuscript.

6. This study pulls from datasets that have already been published by the same group in two prior papers. While there are clearly new observations in this analysis, some of the analytical strategies appear to mimic those used in the prior papers. It is not immediately clear how this study will enable new hypothesis-generating analyses from the community beyond what was enabled from the prior two publications.

Please see our comments above. We did not simply reanalyse data that were already acquired by our laboratory but instead acquired 1.16 TB (terabytes) of completely new mass spectrometry data for this study including all of the phosphoproteomic data. The new NCI60 data is of much higher quality than that of the previous publication and the phosphorylation data enabled layers of data analysis that were not possible before (including e. g. the ATLANTiC scoring system).

Referee #2: bioinformatics, outcome associations, breast cancer (Remarks to the Author):

The paper “ATLANTIC:...” by Frejno et al uses comprehensive protein abundance and phosphoproteome data, in conjunction with drug-sensitivity screens to generate a community resource consisting of landscapes of pathway and kinase activity across hundreds of cell-lines, which can be interrogated via an interactive website (ATLANTIC) to identify patterns of activity underlying drug sensitivity or resistance. This important resource can be seen as a powerful hypothesis-generation tool, and the paper showcases a number of different examples, some of potentially great clinical significance, notably the finding that Adenylate kinase isoenzyme 1 (AK1) inactivates antimetabolites like Cytarabine, and that AK1 levels can stratify Cytarabine-treated AML patients into subgroups exhibiting very different survival.

We are very thankful to the reviewer for their accurate and overall positive summary of our study.

However, I do have a number of major concerns. One major concern is the mentioning *in the abstract* of the stratification of PGR+ breast cancer patients into different prognostic subgroups based on PGR phosphorylation, when the obtained P-value ($P=0.012$, Fig.5G) is only marginally significant. I consider this a major concern, because the authors imply in their manuscript that thousands of valuable hypothesis could be generated from ATLANTIC, yet is a P of 0.012 the best the authors could do here? A P-value of 0.012 can't be regarded as evidence that PGR phosphorylation would stratify PGR patients according to different survival, as such a marginal association could easily be driven by residual confounding. Therefore, further validation of this result in an independent cohort is warranted. Alternatively, the authors could perhaps find another more convincing example of phosphorylation levels stratifying cancer patients into different survival subgroups.

We agree that the obtained P-value may appear weak at first sight and is not as low as for example the P-value associated with the increased survival probability of PGR+ compared to PGR- breast cancer patients. However, we would like to point out that we further (!) stratify the PGR+ group of patients with favorable outcome into two subgroups based on the phosphorylation status of PGR. Detecting a subgroup of patients with worse outcome in this group of patients with overall favorable outcome is much more difficult than the aforementioned stratification according to the expression of PGR protein without pre-selection of patients. When put in this context, the obtained P-value is in fact not weak. However, it is possible that this stratification is confounded by other risk factors. In order to investigate whether this is the case, we performed Cox Proportional Hazards Regression (COXPH) on the breast cancer cohort. We modelled survival probability as a function of pPGR status in combination with various potentially confounding factors such as Age and HER2 status. The results of this analysis are now part of Supplementary Table 9, which highlights that high phosphorylation of PGR (pPGR+) is still a significant predictor of overall survival when combined with the HER2 status of the patients and when combined with the grading of the tumour. However, it is only marginally significant (P-value = 0.056) when combined with the age of the patient or the T grading scale of the tumour (P-value = 0.102). This stresses that while there are confounding factors of our PGR-phosphorylation-based stratification in the context of PGR protein positivity, they are restricted to a ubiquitous confounder (age) and one that itself is not significant. We agree that this might warrant further validation in an independent cohort. We have added a note on this to the manuscript as the authors feel that doing so would be beyond the scope of this already ambitiously expansive study.

Apart from this major concern, I also found this paper very difficult to read. At various places in the manuscript the authors do not clearly explain how the specific analyses were done. While

details are provided in the Online Methods section, the text in Results section is often inconsistent with what is described in Methods, and therefore could be improved.

We agree that the paper might have been difficult to read in places because of space constraints of the original manuscript type. We have restructured the revision to some extent and made sure to elaborate more on how the specific analyses were done in the main part of the manuscript.

I also found the manner in which some of the results in the figures are displayed (e.g. the hive plots in Fig.2) to be overly complicated, which I believe many readers would find “off-putting”.

The authors agree that hive plots can be complicated to read. Still, we believe they are easier to understand than the ‘hairballs’ which are usually used to depict network structures like the one in Figure 2C. With a data density this high, it is nearly impossible to find a simple representation that captures the process in its entirety. Hence, we specifically chose hive plots because of their structured appearance. Figure 2C was also meant to give an overview of the guilt-by-association approach we took in order to place proteins and phosphosites of unknown function into a functional context – an important aspect of the overall story.

Some of the results (e.g. those shown in Fig.2B) are also hard to glean and don’t seem to be in line with what is being said in the main text.

To improve clarity, we generated two additional panels Figure 2C and D, which show the relative activity of kinases in these two cell lines using a waterfall-like plot. We think that this representation makes it much clearer that ABL1 and ALK are highly active in K562 and SR cells, respectively.

Finally, I have a number of major concerns regarding statistical significance estimates generally, as some analyses were not adjusted for multiple-testing- a major unacceptable omission, as it is therefore unclear how many of the thousands of hypotheses that ATLANTIC is supposed to deliver are actually statistically meaningful.

We decided not to adjust for example the correlation-based P-values for multiple testing because we merely used them to rank proteins/phosphosites rather than using them as hard cutoffs. This is because the primary focus of this analysis is to provide a resource that can be queried by scientists interested in specific hypothesis. In most cases, readers will only be interested in one or a few specific associations between drugs and proteins/phosphosites. Therefore, we decided to accept more false-positives rather than generate an inflated number of false-negatives by multiple-testing adjustment. We included parts of the statement above in the Supplementary Methods in order to be transparent and to explain our choice.

Also, the implementation of elastic net and random forests (RF), which only used leave-one-out CV could be problematic, as these algorithms include a number of parameters that need to be tuned in a *nested* cross-validation procedure, otherwise some overfitting may have occurred. Although e.g. the RF-model was tested on an independent panel of cell-lines it remains to be seen if a model trained with nested CV on the CRC panel would also obtain the same R value.

We did not use nested cross-validation during the training of elastic net models but a bootstrapping approach in order to obtain empirical estimates of feature stability. We note in this context that the elastic net models were not used to predict drug sensitivity but rather to find groups of proteins/phosphosites, which are correlated with drug sensitivity. Given that we don't use them for inference, the reviewer's concern regarding potential overfitting of elastic net models is not

justified. Regarding the random forest (RF) models, we point out that the models were trained on the CRC65 dataset and tested on the NCI60 dataset (see Supplementary Methods). Since the models never ‘saw’ the NCI60 dataset during training, we argue that it is not important for our conclusions whether or not nested cross-validation was used during training. If overfitting would have occurred, the models would not perform well on the colorectal cancer cell lines of the NCI60 test dataset but they do. Therefore, we can exclude that the RF models suffered from severe overfitting.

Below I list the main concerns:

Major concerns:

1) PGR phosphorylation stratifying PGR+ breast cancer patients: The P-value in Fig.5G is only marginal, and therefore this result needs to be strengthened by demonstrating a significant association in an independent cohort. Also, please add a measure of prognostic separability (e.g. HR or D-index) and 95% CI to these KM plots.

Please see our comments above. Considering the measures of prognostic separability, we now show the Hazard ratio in the KM plots in the new Figure 6F.

2) Hive plot in Fig.2C: I did not find the hive-plot very illuminating, but rather confusing to say the least. In fact, can't the same information be shown as a series of color bars arranged as a heatmap? I think so....therefore, as it stands the hive plot is an unnecessary waste of space and may only confuse readers.

Please also see our comments above. The same information can in fact not be shown as a series of color bars arranged as a heatmap, since Proteins and p-sites (rows in a potential heatmap) can

be part of multiple GO-terms or KEGG pathways. Therefore, the information contained in the first axis of our hive plots (Groups of functionally related Proteins/p-sites) would require a separate column of bars for each GO-term and KEGG-pathway, which is impossible to show in a figure because there are thousands of different functional annotations on this axis.

3) Fig.2B: This panel is used to show associations for the K562 and SR cell-lines, yet the associations mentioned in the main text are hard to discern in the heatmap. I think the authors need to display this in a more convincing way. This is extremely important, because the authors never validated or benchmarked their method for estimating pathway activity. I would have liked to have seen some benchmarking or clearer validation in this figure.

Please also see our comments above. Considering the benchmarking or validation of the results in this figure, we point out that known genetic alterations were recapitulated by our scoring, which validates the usefulness of our approach.

4) Lines 71-72: It was unclear to me what exactly the authors did when trying to assess if the biological pathways are “recapitulated” in their data. Later in Online Methods it transpires that they assessed this by first identifying significant correlations between proteins or p-sites and then checking whether these are enriched for interactions contained within biological pathways. It is very important that the authors clarify what the analysis entails in the main text. One sentence is sufficient!

The authors agree that this particular analysis was not sufficiently outlined in the main text, which is why we added a sentence describing it to the relevant section.

5) Hypothesis-free vs Hypothesis-driven: the way this section was written should be improved, because there is no logical reason why elastic net was applied to the hypothesis-free analysis and not to the hypothesis-driven one. Elastic net and RF are just two different types of machine learning methods (one is linear, the other is not), and one is left wondering what the result of Elastic Net on the hypothesis-driven analysis is? Is the use of RFs necessary to get a significant R-value in the NCI panel? If yes, this is important to know and the biological significance of the resulting trees should be discussed.

We agree that this section was not clearly written, which is why we rephrased it and excluded the distinction between hypothesis-free and hypothesis-driven analyses.

6) Code availability: it would appear that the authors are not making their code available and only upon request, but this is not acceptable. All code used to generate the main results of the paper should be made freely available in a way that others can easily reproduce the findings.

We are in the process of making the code available on github here <https://github.com/kusterlab/ATLANTiC> and will help our readers to reproduce the analyses presented in the current study if the need arises. The code will be available before the anticipated publication of the ATLANTiC manuscript.

7) Lines 170-172: This part of the paper is poorly presented. The authors state that they used elastic net and random forests for hypothesis free and hypothesis-driven approaches, without specifying what on earth the hypothesis-driven approach is. As it stands this sentence implies that RFs is designed for hypothesis-driven testing whereas elastic net is designed for hypothesis-free testing, which is completely untrue. These are just different machine learning methods, one is linear and the other is not, which can be BOTH applied in hypothesis-free or hypothesis driven settings. In

the next sentence, the authors report millions of protein-drug and p-site-drug associations, which we presume are statistically significant, but the statistical significance level is never mentioned.....moreover, from the text we would think that the statistical significance seems to have only been assessed in a multivariate (elastic net) setting, yet in Online Methods the authors report a “Simple Correlation Analysis” where P-values were not even adjusted for multiple-testing: it is not acceptable to quote millions of associations without even correcting for multiple-testing.....

Please see our comments above.

8) Statistical Significance of correlations is unclear: Related to the previous point, and also relevant for say Extended Data Fig.2, it is very unclear over how many cell-line samples correlations were estimated given that the amount of missing data was substantial. For instance, I checked the ATLANTIC website for the drug Cladribine, and did a simple correlation analysis. I then picked a number of top hits, and for many, correlations appear to have been estimated over only 9 or 10 cell-lines. Since the authors are computing large numbers of drug – protein/p-site pairs, yet most of these are only assessed over 10 or so cell-lines, a very large number of false positive associations could occur. Perhaps, the authors could provide a distribution displaying the fraction of drug – protein/psite pairs that were evaluated over at least x number of cell-lines with x varying between 7 (the minimum number considered) and the maximum value (the number of cell-lines in the panel).

Some of these points were already addressed in earlier comments of ours, but we would like to emphasize the key points here again for clarity. We refrained from adjusting the correlation-based P-values for multiple testing, since we merely used them to rank proteins/phosphosites. Adjusting

for multiple testing only controls the false discovery rate when considering the entire dataset, but that's not the main use case here. The ATLANTiC web portal is meant to help readers explore our data with a specific hypothesis in mind. This means, a reader will have a drug or protein/phosphosite in mind, which they want to find in our dataset. In the majority of all cases, they will not be interested in the associations between all proteins/phosphosites and the sensitivity profiles of all drugs. Therefore, we decided to accept more false-positives rather than generate an inflated number of false-negatives by adjusting our correlation-based P-values for multiple testing. Regarding the number of missing values during the calculation of these correlations we note that the P-value of a correlation is not only dependent on the strength of the correlation but also on the degrees of freedom during the calculation of said correlation. In other words, the P-value of a correlation also depends on the number of cell lines that were used during the calculation of the correlation. If fewer cell lines were used, the P-value of a correlation of a given strength is higher. When we rank the correlations first by P-value and then by the actual correlation coefficient, we ensure that the most reliable correlations appear at the top of the list. In the ATLANTiC web portal, users can inspect the scatter plot of $-\log_{10}(P\text{-value})$ vs correlation coefficient in order to get an idea of which correlation is the most reliable. We believe that this plot is actually superior to the one suggested by the reviewer when assessing the reliability of correlations.

Referee #3: computational, proteomics data (Remarks to the Author):

ATLANTiC: Activity landscapes of tumor cell lines determine drug responses

Summary: The authors present a study of proteome activity landscapes of cancer cell lines, which they use to assess functionalized proteins and phosphorylation sites, and predict drug sensitivity. Overall, the work analyzes 125 cancer cell lines for 10,000 proteins and 55,000 phosphorylation sites, characterizing phenotypic drug sensitivity for ~900 drugs and molecular drug target selectivity information for 224 drugs. From this analysis, the authors make two main conclusions: (1) progesterone receptor phosphorylation can be used to stratify breast cancer patients into those with short or long survival times; and (2) the enzyme adenylate kinase isoenzyme (AK1) inactivates cytarabine, a standard chemotherapy for acute myeloid leukemia (AML), making high AK1 a good biomarker for poor survival in AML. As a resource for the community, the authors provide their findings in an interactive website where the role of single proteins and drug sensitivity can be analyzed online.

Major Points: This work's strengths lie in the comprehensive analysis of hundreds of chemotherapies, and identification of proteins and phosphorylation sites affected by these compounds. The dataset and interactive website provide a unique, valuable resource for cancer biology studies, and moreover for drug development. The combination therapy in vitro assays that capitalize on the outlier phospho-sites support the utility of this work. Furthermore, the computational correlation-based analyses present plausible phosphoproteome markers to explain drug sensitivity, and the application of the sparse multiblock partial least square regression analysis to identify common phenotypic drug sensitivities is elegant. However, while the phosphoproteome data, drug sensitivity analysis, and website seem meritable for a short resource publication on their

own, the translational conclusions of the work seem premature: use of cell lines pose a major limitation to the ability to translate these findings clinically. The cell proteome from patient biopsies would be anticipated to be significantly different from cell lines, as has been shown (Hu et al., 2019, Nature Biomedical Engineering; van Galen et al, 2019, Cell).

We agree that cell lines often fail to mimic a patient's tumor but this is not universally the case (e.g. for Her2 positive breast cancer where several cell lines recapitulate the disease biology and which are still heavily used in the pharmaceutical industry). In addition, cell lines are still indispensable preclinical models, particularly for understanding the effect and mode of action of anti-tumor drugs. Publically available major drug screens on e.g. 3D models or PDX models do not exist. This may change over time but for the time being, the field still relies on the use of 2D cell lines and the extensive amount of molecular and phenotypic data available for these models. Therefore, studies from the very recent literature investigating cell lines are still of high interest to the scientific community (e. g. Ghandi et al., Nature 2019; Li et al., Nat Med 2019).

Representing a wide spectrum of genetic backgrounds, activated pathways and tumor subtypes requires the analysis of a large number of cell lines. Therefore, we believe our study of more 125 cell lines (the largest proteome and phosphoproteome panel to date) provides a valuable starting point for researchers to develop their own translational hypotheses together, of course, with many additional lines of evidence. In this study, we could only highlight two such examples, demonstrating the translational value of our resource. More specifically, the AKI finding is not only interesting from a drug MoA point of view, it is clinically actionable. As a straightforward clinical translation example, our colleagues in oncology are already considering to use AKI measurements to adjust the dose of cytarabine given to patients.

The claims for clinical / in vivo relevancy are moreover weakened by specifics of how the patient tissue samples were analyzed as described below (Points 2-5).

Please see our comments below.

Point 1: p. 5, The authors present a new computational analysis to identify interactions of the phospho sites and proteins, and differences across cell lines. However, minimal validation of known interactions is presented (while a few citations are provided, no experimental validation is performed on the functional interactions described in this section).

Please also see our comments to other reviewers above. We acknowledge that we did not validate the functional interactions discovered in this particular section because we placed the emphasis on hypothesis validation on the 'translational' cases of AKI and PGR. Additionally validating some of the hundreds of associations between phosphosites or proteins and drugs would, therefore, seem beyond the scope and length of this manuscript. In addition, for this resource paper, we expended a lot of effort to make our data available (in terms of data deposition) and accessible (via the web site atlantic.proteomics.wzw.tum.de) to the scientific community in order to promote follow-up experiments or alternative types of (re-) analysis beyond the few validated showcases that we were able to fit into one manuscript here.

Point 2: In immunostaining for the PGR+ breast cancer patients, any nuclear expression of PGR in a patient tissue sample led to a positive classification. The samples seem to be taken at diagnosis / surgery and not following response to therapy. The qualitative nature of the PGR+ label along with the skewed distribution of the PGR- (84 patients) vs. PGR+ (277 patients) and the use of overall survival probability as a clinical outcome rather than remission duration or an assessment

of the response to endocrine therapy, all weaken the claim that the analysis ‘predicts benefit of endocrine therapy in breast cancer.’

The authors point out that the current work was not part of a clinical study asking the question if pPGR predicts benefit from endocrine therapy in breast cancer. The patients were part of a (retrospective) cohort who were diagnosed with breast cancer and received surgery between 2004 and 2012 (stated in the Supplementary Methods). The qualitative staining and classification with respect to PGR used in this study is the standard procedure performed at diagnosis. There was no further pre-selection of the patients based on hormone receptor status, which explains the skewed distribution of PGR+ vs. PGR- patients. Instead of manually picking a subset of PGR+ patients, we decided to rather use all data collected in order to avoid biasing the results. We respectfully disagree with the reviewer that the skewed distribution of PGR+/- patients weakens the claim that pPGR predicts benefit of endocrine therapy, since we only analyzed PGR+ patients to come to that conclusion. Since all PGR+ patients received endocrine therapy, we refrained from using remission duration as the endpoint but instead used overall survival. We updated our KM plots and adjusted the corresponding subheading of the results section to make this clearer.

Point 3: Similar to the breast cancer analysis, the authors present immunostained data for AK1 bone marrow biopsies for AML patients correlating high or low level AK1 to survival, claiming that this “strongly supports” AK1 can dephosphorylate nucleoside analogue drugs in vivo. However, the levels of AK1 could be indicative of patients that would have very low overall survival, independent of their response to cytarabine. Moreover, the criteria for defining high vs. low AK1 expression is subjective (scored by 2 pathologist at 0, 1, 2, 3, where 0 and 1 are low and 2 and 3 are high); this could be mean that bone marrow samples that are close in expression (e.g., scores of 1 and 2) were separated. Also, the skewed distribution of the AK1 high levels (21

patients) vs. AK1 low levels (58 patients) and the use of overall survival as a clinical outcome rather than remission duration or immediate response to cytarabine weakens the assessment.

Please also see our previous comment. We note that the work was not part of a clinical study asking the question if AK1 levels predict benefit from cytarabine treatment in AML. Clinical studies often display an imbalance in the number of patients with a specific genetic background. For example, only 15-20% of breast cancers are triple-negative (Kohler et al., JNCI 2015), but these patients react dramatically different to endocrine therapy compared to other subtypes of breast cancer. Therefore, we believe that the evaluation of whether AK1 is a useful biomarker to stratify patients in the clinic should not be influenced by the number of patients in each of the groups. In addition, we point out that all patients in the AML cohort received at least one course of cytarabine (see Supplementary Methods). Similar to the breast cancer cohort described above, we therefore refrained from using remission duration as the endpoint but instead used overall survival.

Suggestions to strengthen the claims of the manuscript prior to publication would be to (1) extend the AK1 overexpression + cytarabine treatment from just Jurkat cells to AML patient cells from blood or bone marrow, and (2) assess immediate response to cytarabine therapy or remission duration (i.e., how many patients when into remission and/or how many stayed in remission). Alternatively, rather than “strongly suggesting” that AK1 can dephosphorylate nucleoside analogue drugs in vivo, can in vivo studies be performed?

Unfortunately, the information about remission duration and the corresponding response measurement is not available. We also think that measuring the dephosphorylation of nucleoside analogue drugs in vivo is beyond the scope of this resource manuscript.

Point 4: What is the negative control (normal) for the comparisons to the AML bone marrow histology samples stained for AK1?

We have validated the antibody in AK1 knock-out cells prior to the IHC staining. We added a sentence to the Supplementary Methods in order to clarify this.

Point 5: Would the authors clarify whether the 79 bone marrow samples were taken at diagnosis only (that is, before therapy)?

Yes, the samples were taken at diagnosis and before any therapy. This information was already provided in the Supplementary Methods. We have added a sentence on this to the main manuscript.

Point 6: p. 13 The conclusions of the manuscript, that the activity landscapes of tumor cell lines can be assessed by the authors' methods, and used to predict the response of cancer cell lines to in vitro chemotherapy seem strong, while the 'extremely high translational value' claim does not. Suggestion to either change this emphasis in the Conclusions or perform additional in vivo or patient cell biopsy experiments backing up the translational value.

Please also see our comments above. The authors still think that our results have translational value but agree that this may have been overstated. Therefore, we have toned down these statements in the revised manuscript.

Minor Suggestions:

Overall, the interactive website is nicely designed, fast and functional. However, there are some pages that would benefit from a redesign, as they are not intuitive or easy for the user to navigate. As one example, when a drug is selected and a simple correlation is presented, the user needs to

navigate the Volcano plot blind to identify drug protein/phospho-site associations when it would be helpful to include the ability to look for phospho-sites in a sub-search or include their labels in a “hover-over” of each point on the Volcano plot. The current design also seems directed towards a proteomics audience – to expand its utility, a video or interactive tutorial would benefit the audience.

We are in the process of preparing an extensive tutorial to help users navigate and explore the website more easily. This will be available on the ATLANTiC web portal before the anticipated publication of the ATLANTiC manuscript. Volcano plots in the ATLANTiC web portal can be manipulated in several ways. We support zooming in/out, adding different labels and highlighting specific subsets of data in different colours. In addition, the selected proteins/phosphosites/drugs are listed at the bottom of the page in tabular form.

p. 1, Check and standardize numbering and affiliations for authors (e.g., as is “Chair of Proteomics...is associated with two authors; should “current address” still be in front of some affiliations but not others?)

We have cleaned this up in the revised manuscript.

p. 2, line 42, Add commas to guide the reader: “...by, e.g. dynamic phosphorylation,”

We refrained from implementing this change because we feel that it rather confuses a reader.

p. 2, line 44, Add comma before the and: “...drugs, and some...” This sentence merits citations.

We have changed the sentence as recommended and included two citations.

Referee #4: proteomics (Remarks to the Author):

This study by Kuster and colleagues presents an integrated proteomic and phosphoproteomic analysis of ~125 cancer cell lines correlated with drug sensitivity and pathway analyses. The core of this work is the extensive label-free quantification of bulk proteome and phosphoproteome across the NCI60 and CRC65 cell line panels. Separately, phosphopeptide enrichments were employed to detect and quantify phosphorylated sites and proteins across these cell lines. The resulting dataset and pathway level data have been integrated with existing drug sensitivity resources (i.e. previously published and public datasets) to identify correlations between protein and p-site levels with drug mechanism of action and sensitivity. The authors detected upwards of 10,000 protein groups and tens of thousands of phosphosites across this panel, and were able to synthesize this into functional enrichment and gene ontology analyses across cell lines. As a resource the large integration of data that likely only partially exists in separate studies and under different conditions will be useful to the community through the primary data and online portal that the authors have created. The authors then show that known therapeutic liabilities can be detected using the dataset and integrated drug sensitivity data, for example with Bcr-Abl inhibitors and arsenic trioxide. The final two examples presented demonstrate the ability to identify cancers that may be uniquely targeted by specific drugs as a result of their harboring high protein abundance, phosphorylation, or pathway activity. This is specifically explored with PGR and AK1, which are correlated with drug effect and metabolism, respectively.

Central to whether this manuscript has far reaching impact and provides new insights into the mechanisms underlying cancer cell phenotype and drug sensitivity is the focus on identifying pathways that are 'outliers' or enriched in certain cell lines and lineages, and whether these can highlight novel therapeutic insights. Overall, I think the manuscript is confusing and falls short in

this respect. The pathways analyses rely on opaque definitions of up- and down-regulated protein groups and integrated phosphoprotein categories that even upon reading the supplement in great detail are still not obvious in some cases. Additionally, standard gene ontology definitions from existing databases are relied upon to extract meaning in these enrichment analyses. These terms and definitions are notoriously biased and flawed, and the true utility of a study such as this is that new insights could be made without relying on these current databases. In fact, the major issue with the utility of the current dataset is that the depth of pathway analysis and drug sensitivity discovery is quite shallow compared to previous large scale studies/resources that have done the same things but with gene expression data (e.g. the connectivity map, CTD2, and various studies with CCLE, TCGA et al.). There is not strong case that the protein level and phosphoprotein level data are enabling for the discovery of dependencies at a level that is beyond those previously published and the gene expression level, and many of the novel dependencies presented have significant precedence for their existence already. Based on these factors and more specific details included below, it's not clear how this study can add to our mechanistic understanding of cancer drivers and drug sensitivity in its current form.

Please see our detailed responses below. Naturally, the authors do think that this work is important and goes far beyond the prior literature. Please also see our comments to other reviewers above. More specifically, as most drugs act on proteins and given the many uncertainties in extrapolating protein levels from transcript levels, measuring protein levels directly is valuable per se. The same can be said from measuring phosphoproteome levels as this information cannot at all be deduced from transcriptomics. As an example, transcript and proteome levels of MET and MST1R (or RON) would not have turned up anything interesting with respect to drug sensitivity as their response to certain drugs is solely governed by their phosphorylation status.

Significant concerns:

1) -The integration of multiple streams of quantitative data to generate kinase pathways scores is interesting and could be a useful way to identify unknown connections between drug sensitivity and targets or synthetic lethal liabilities in the cancer cell lines studied. The method in which these pathway scores were calculated, however, is likely to bias toward only the most well studied kinases and their daughter networks such that pathways/kinases known to be highly active in a specific context are just reprioritized and rediscovered. This is because by integrating kinase abundance/phosphorylation and substrate pathway member phosphorylation you are inherently providing much higher data quality and depth for known targets (i.e. many substrates proteins and sites known) vs. less well-studied targets that have few known sites and substrates. Thus, the ATLANTIC workflow touts itself as a global and unbiased discovery dataset and tool, but in the end it relies on known connections and existing annotations that are biased toward the small fraction of kinases and p-sites that have some degree of characterization. Indeed this appears to true in the analysis in Figure 2, where known pathways are highlighted, and no new insights are made, or at least discussed. Can the authors alter this analysis to bias away from what is already known? There are several other examples where these existing, surface level annotations are relied upon, such as in figure 1E, resulting in the same issues.

The authors felt that it is important to show that the data can actually recapitulate known biology before venturing into new space. It is indeed true, that we cannot identify unknown connections between drug sensitivity and targets using our ATLANTiC scoring scheme, which is the focus of our landscape plots in Figures 2A and B. This approach was meant as a concise and in our opinion intuitive way to show highly active pathways or kinases in subgroups of cell lines based solely on known annotations. In addition, we believe that these plots are useful for showing how

heterogeneous and diverse the cell lines of the NCI60 and CRC65 panels are. We changed the appropriate paragraphs of the manuscript to reflect this.

While the landscape plots were biased towards known kinases and their annotations with a focus on characterizing the cell lines, we devoted a different analysis to the discovery of unknown relations between proteins/phosphosites. We employed a guilt-by-association approach in Figures 2E and F, as well as in Supplementary Figure 3C, which placed many of the so far uncharacterized phosphosites into a functional context. We made the results of this analysis available through our ATLANTiC web portal, which enables users to explore them dynamically. This is an example where we used known annotations in conjunction with a correlation-based analysis of our quantitative data in order to reveal previously unknown relations between proteins/phosphosites. Using pathway and GO-term information is widely accepted in the field, both in a more general form of gene set enrichment analysis (Subramanian et al., PNAS 2005, 15000+ citations) and in studies closely related to the scope of our manuscript, i.e. identifying tumor-driving kinases/pathways in cell lines (Beekhof et al., Mol Syst Biol. 2019). While we are aware of the fact that these annotations are notoriously biased towards well-studied proteins, we believe that we have found a way to harness that information in order to gain new insight into poorly characterized proteins/phosphosites.

In addition, we have applied several other methods to analyse our data and answer different questions. For example, we used simple correlation analysis to find which individual proteins/phosphosites are correlated to a drug response, we used elastic net regression to find single and potential combinatorial markers predicting drug response and we used SMBPLSR to find how common markers for drugs with similar drug responses.

2) The terms “enriched”, “decreased abundance” and “increased abundance” are used pervasively throughout the manuscript. What is never actually explained is what exactly is being measured (for example with some p-site definitions), are changes absolute or relative, and on what scale these changes are occurring?

The authors apologize if this was unclear. We defined up- and down-regulation, as well as high and low abundance in relative terms without specifying any metric or scale when comparing the expression of one protein over different cell lines. To make this clearer, we have added additional text to the ‘Data post-processing and filtering’ section of the Supplementary Methods as follows: ‘If not stated otherwise, all comparisons throughout this manuscript are relative. If we state that a protein/phosphoprotein/p-site is ‘high’ in a certain cell line, then it means that it’s abundance is higher compared to other cell lines without specifying any further metric or scale.’. Whenever possible, we show all the information in scatterplots where the x-axis shows protein abundance and the y-axis shows drug sensitivity. From these plots, the dynamic range of the protein expression, drug sensitivity changes and the correlation of the two quantities are completely transparent to the readers. Whenever such a plot was not possible or impractical, all the information can be retrieved from the ATLANTiC web portal. In this study, we have explored the association between tens of thousands of proteins/phosphosites with 900+ compounds. Simple criteria like fold-changes or P-values do not fully capture the quantitative nature of these protein/phosphosite and drug response readouts. We have overhauled the text in an attempt to avoid excessive repetition of the same terms. Using consistent terms is however necessary in order not to confuse readers.

For example, some statements in the text suggest that the overall phosphoprotein intensity is being drawn from multiple sites, and this is confirmed by the statement in the methods “In addition to

our p-site data, we also rolled-up MS1-based phosphopeptide intensities to form phosphoprotein intensities analogous to what is commonly done for peptides to calculate protein intensities.” Integrating phosphopeptide intensities across a protein to give a measurement of overall “protein phosphorylation” is highly unusual and misleading. Each phosphosite is related to unique proteoforms, they are not necessarily caused by the same kinase(s), involved in the same pathway(s) and certainly do not have the same effect(s) on protein function. Combining them overlooks all of these features and dilutes any interpretation at best, and completely misdirects them at worst.

We fully acknowledge that integrating the intensity of multiple phosphopeptides to calculate aggregated “protein phosphorylation” values can be misleading. Therefore, we actually explored aggregated and non-aggregated abundance data for phosphoproteins and individual phosphosites in this work. As far as aggregation is concerned, we and others see potential in this approach. For example, it has been frequently observed that some biological functions of signalling proteins are fine-tuned by multiple phosphosites simultaneously (e. g. Witzel et al., Biophysical Journal 2018). Hence, looking at single phosphosites may fail to capture this interesting biology. In addition, such aggregation approaches are not as unusual as it may appear. Very similar concepts (using the average intensity of multiple phosphosites from the same protein) have been successfully applied in other studies (e.g. Vasaikar et al., Cell 2019 and Mertins et al., Nature 2016). The authors are fully aware of the limitations of this approach. We expect that each level of quantification has advantages and disadvantages. Examining all of them as we did in our manuscript can provide complementary views on biological mechanisms.

These details need to be clarified explicitly, such that the species being measured (phosphosite or integrated phosphoprotein or other) are clearly explained in the text, and moreover that the

definition of “increased” and “decreased” throughout the text needs to be more clearly defined in each case.

We checked the text again to make sure that the entity being measured is clearly stated in the text.

3) The authors stated early on that the level of phosphorylation for many proteins was strongly correlated with overall protein abundance. In figure 5 the authors show that PGR- patients (at the protein level) exhibit worse prognosis overall and that a similar trend is observed for p-PGR. This trend is less pronounced, and is it not just a further sub-analysis of the protein level effect? Moreover, it is well established that progesterone receptor negative breast cancer patients have worse prognosis, partially because it includes patients that are triple negative. It's not clear that anything new is being uncovered here, or that the phosphorylation level analysis is providing any new information. Can the authors clarify these points and highlight what is novel in these biological annotations?

Please also see our response to reviewer #1. There is a misunderstanding that arose from the fact that the data we showed in Figure 5F and G were not clear enough. We now combined them into a single panel in Figure 6F. We believe that this panel makes it much clearer that PGR+ breast cancer patients who were pPGR- have a lower chance of survival than patients who were PGR+ and pPGR+. We do not compare to PGR negative cases here.

4) Some of the early correlations and conclusions about pathway level enrichment are very confusing in the text. For example, what is figure 1E showing? The text casually mentions associations of various functional categories, but in what cell lines, with what treatments? What comparisons are being made? Is this referring just to protein abundance level data? Or phosphoprotein data? It is very hard to follow or derive any meaning from these plots.

This information was previously in the Supplement due to length restrictions on the initial submission. We have now extended the explanation of this analysis at the appropriate position in the revised manuscript and hope that it is now clearer.

5) Can the authors provide compelling data that shows that the phosphorylation level data provide additional insights that cannot be found with gene expression or protein level data alone? It's not a problem if they all correlate of course, but a major advantage of this dataset is the ability to query the phosphorylation level information. It's not clear that this is clearly demonstrated in the current manuscript.

The authors apologize for not bringing this out more clearly. Examples showing the value of the phosphorylation data are provided in several places of the manuscript. First, in the part of the manuscript systematically exploring which protein-protein interactions can be recapitulated at the protein or phosphosite level, the results of which are shown in Figure 1E and Supplementary Figure 2. The results are discussed in the manuscript and we concluded that dynamic processes such as the cell cycle are better recapitulated at the level of phosphosites, whereas constitutive processes such as spliceosomal assembly are better captured by the proteome data (Supplementary Figure 2). Second, we not only placed proteins but also phosphosites into new functional contexts using guilt-by-association, thereby taking the first steps towards functionalizing hitherto uncharacterized phosphosites (Figure 2E and F). Third, we showed that the phosphorylation status but not the protein level of MET and MST1R/RON suggests that Tepotinib and MK-2461 might show synergy in HDC-8 cells, which we validated using additional in vitro experiments. Fourth, we exemplified in Supplementary Figure 4 how different levels of omics data can lead to complementary information in specific cases. Finally, we note that the additional stratification of PGR+ breast cancer patients based on the phosphorylation status of

PGR was only possible because of our quantitative phosphoproteomics data. Taken together, we believe that we highlighted several cases that show the value of acquiring phosphoproteomics data alongside genomics, transcriptomics and proteomics data in order to capture a more comprehensive molecular picture of cancer cell lines.

Reviewers' Comments:

Reviewer #1:

Remarks to the Author:

thank you for the thorough responses to my initial critiques. i have no further comments.

Reviewer #2:

Remarks to the Author:

In the revised version of the paper the authors have addressed a few of the major concerns, but unfortunately some of the key concerns remain. I fully understand that the authors have generated what appears to be a very comprehensive dataset with a very valuable webserver, and therefore this should in principle constitute a very valuable resource to the cancer and drug communities. However, this in itself should not give the authors licence to overinterpret findings and to convey wrong or misleading impressions. While the authors imply that this resource is a gold-mine for generating many novel hypotheses, I insist that it is therefore very disappointing that in relation to the phosphoproteome analysis the most convincing example that the authors come up with is a very marginal association in stratifying PGR+ patients based on pPGR status. Moreover, I did not find the author's response very convincing. First of all, adding the comparison now in relation to the PGR- group is not really helpful, as my concern relates to the stratification of the PGR+ patients into pPGR+ and pPGR-. I indeed see that there are only 41 PGR+/pPGR- samples, and I count on the order of only 5-6 events in this group, so I can see that the authors are somewhat underpowered. But this is precisely also why a validation in an independent cohort is absolutely necessary. In fact, if there is one lesson that we should have learned from the two decades of omic data analysis is this: always validate your findings in an independent cohort, because marginal associations could easily be driven by residual confounding. While the authors do attempt to address potential confounders (age, HER2+ status, T grading), the analysis adjusted for T-grading renders the association now not significant ($P=0.1$), indicating that indeed the association in the unadjusted analysis may well have nothing to do with the phosphorylation status of PGR. In my sincere opinion, if the authors want to mention in the abstract that pPGR can stratify PGR+ patients into two prognostic groups, it is imperative that the authors add an independent validation cohort. Otherwise, please remove any mention of this in abstract, discussion and conclusions, because it is utterly misleading. If the phosphoproteome data is not helpful in identifying any other prognostic subgroups in other cancer-types then this negative result should in fact be mentioned. Personally, I think that the community benefits more from an honest account about the value of the phosphoproteome resource (or perhaps I should say the lack of value in relation to clinical outcomes) rather than hyping up a very marginal and very possibly meaningless result as if this was somehow just one example of the many other associations with clinical outcome present in this database. Where are these other examples? Is this gold-mine full of gold or just empty? In the revised version the authors have not added any further examples, nor have they added an independent validation to support the preliminary pPGR finding.

Besides this major concern with the pPGR-analysis, the other major concern which the authors have not addressed is the fact that they have computed millions of correlations without ever adjusting for multiple-testing. Once again, I did not find the author's response to be convincing, because there are many different statistical methods available to adjust for multiple testing, some very stringent and conservative (such as Bonferroni) and others which are less stringent (such as say using an $FDR < 0.05$ or even < 0.3 threshold). So, when the authors state that they did not adjust for multiple-testing because they worry about large numbers of false negatives, they should realize that this only applies to Bonferroni-type adjustments, and not to the FDR. Indeed, the FDR was invented precisely in order to have increased power (ie less false negatives) while also being able to control the number of false positives. What is not acceptable is to allow potentially large numbers of false positives to be shown on their webserver.

Moreover, in my comments I also asked to see a plot displaying the distribution of the fraction of

drug-protein/psite pairs that were evaluated over at least x number of cell-lines with x varying between 7 (the minimum number considered) and the maximum value (the number of cell-lines in the panel). This could be easily added as a SuppFig, and yet the authors have refused to show this important figure. Without this key figure, we can't be certain how comprehensive this study is, because if say 70% of the correlations were only estimated over 8 cell-lines, this would strongly undermine the claimed strengths of this study.

Finally, full code is not yet available and this should be made available before reviewers assess the manuscript, not upon publication, as in my experience these promises are often not honored.

Reviewer #4:

Remarks to the Author:

In the revised version of this study the authors have performed additional experiments and analyses to support the claims that the ATLANTiC tool and database can help identify pathways and targets that are unique to certain cancers. Overall, several explanations to issues raised by myself are satisfactory to support publication of this resource. There are still several issues that confound the use of this database and analysis tool and interpretation of the data in this paper. Upon addressing these issues I think this manuscript should be published.

1) Chief among these is the lack of clarity about the relative abundance metrics used throughout the analysis of cells lines. The authors state in their supplement in response to questions by several reviewers that "if we state that a protein/phosphoprotein/p-site is 'high' in a certain cell line, then it means it's abundance is higher compared to other cell lines without specifying any further metric or scale."

Since the abundance of sites or proteins, which are hard to separate throughout the manuscript, is the core of this integrated proteomic analysis aimed at uncovering markers of disease and drug sensitivity, how can there be no metric or scale in these comparisons? This implies that there is no statistical basis behind these values. This brings into question the entire analysis and its utility if this cannot be adequately explained by the authors and presented to the readers and ultimate users of this database. There should be some sense of degree of abundance change (1.1-fold, or 100-fold?) as well as confidence in that change based on statistical measures.

2) The authors acknowledged in their response that some treatment of phosphoproteomic data, such as rolling up all phosphopeptide values into an integrated protein phosphorylation value, may be misleading. The explanation for this in the manuscript, which is that this is analogous to what is done with normal peptides for proteins, does not address the issue. The phosphorylation sites on that protein are not all placed by the same kinases, regulated by similar phosphatases, or result in similar effects on the protein or signaling. This needs to be more clearly stated in the manuscript and separated from the specific phosphorylation analyses.

Detailed response to reviewer comments:

The authors thank the reviewers for their comments which we took to heart while revising the manuscript. The authors believe that the resulting new figures, new data analyses and new text have made the manuscript much stronger and are confident that all concerns have been adequately addressed. We briefly summarize the most significant changes in the following bullet list before going into more detail further below:

Summary

- *We removed the claim related to the stratification of PGR+ patients based on pPGR from the abstract, toned down our conclusions and clearly mentioned the need for an independent validation cohort at the appropriate positions in the manuscript.*
- *We added FDR-adjusted P-values to the ATLANTiC web portal, which are calculated on-the-fly and are displayed in the corresponding scatter plots instead of unadjusted P-values. This should enable our readers to better judge the significance of the results we display.*
- *We added a plot displaying the distribution of the fraction of drug-protein/p-site pairs that were evaluated over at least x number of cell lines with x varying between 7 (the minimum number considered) and the maximum value (the number of cell lines in each of the panels). This enables our readers to better evaluate the meaningfulness of the correlation analysis.*
- *We now clearly state the definition of “high” and “low” at the appropriate positions in the manuscript. More specifically, we always refer back to the LC-MS intensity values of the proteome and phosphoproteome data and use fold-change values between the highest and lowest values in order to give readers a sense for the scale and dynamic range of the data.*
- *We added a statement explaining potential problems with our strategy to roll up phosphopeptides to phosphoproteins; i.e. that the phosphorylation of different sites on the same protein may be regulated by different kinases and phosphatases and can result in different effects on cellular signaling. We hope this raises sufficient awareness for this simplification that was necessary for being able to use existing protein and pathway annotation information that is not available at the p-peptide level.*

Reviewer #1 (Remarks to the Author):

thank you for the thorough responses to my initial critiques. i have no further comments.

The authors are happy to read that the revision adequately responded to the concerns raised.

Reviewer #2 (Remarks to the Author):

In the revised version of the paper the authors have addressed a few of the major concerns, but unfortunately some of the key concerns remain. I fully understand that the authors have

generated what appears to be a very comprehensive dataset with a very valuable webserver, and therefore this should in principle constitute a very valuable resource to the cancer and drug communities. However, this in itself should not give the authors licence to overinterpret findings and to convey wrong or misleading impressions. While the authors imply that this resource is a gold-mine for generating many novel hypotheses, I insist that it is therefore very disappointing that in relation to the phosphoproteome analysis the most convincing example that the authors come up with is a very marginal association in stratifying PGR+ patients based on pPGR status. Moreover, I did not find the author's response very convincing. First of all, adding the comparison now in relation to the PGR- group is not really helpful, as my concern relates to the stratification of the PGR+ patients into pPGR+ and pPGR-. I indeed see that there are only 41 PGR+/pPGR- samples, and I count on the order of only 5-6 events in this group, so I can see that the authors are somewhat underpowered. But this is precisely also why a validation in an independent cohort is absolutely necessary. In fact, if there is one lesson that we should have learned from the two decades of omic data analysis is this: always validate your findings in an independent cohort, because marginal associations could easily be driven by residual confounding. While the authors do attempt to address potential confounders (age, HER2+ status, T grading), the analysis adjusted for T-grading renders the association now not significant (P=0.1), indicating that indeed the association in the unadjusted analysis may well have nothing to do with the phosphorylation status of PGR. In my sincere opinion, if the authors want to mention in the abstract that pPGR can stratify PGR+ patients into two prognostic groups, it is imperative that the authors add an independent validation cohort. Otherwise, please remove any mention of this in abstract, discussion and conclusions, because it is utterly misleading. If the phosphoproteome data is not helpful in identifying any other prognostic subgroups in other cancer-types then this negative result should in fact be mentioned. Personally, I think that the community benefits more from an honest account about the value of the phosphoproteome resource (or perhaps I should say the lack of value in relation to clinical outcomes) rather than hyping up a very marginal and very possibly meaningless result as if this was somehow just one example of the many other associations with clinical outcome present in this database. Where are these other examples? Is this gold-mine full of gold or just empty? In the revised version the authors have not added any further examples, nor have they added an independent validation to support the preliminary pPGR finding.

The authors certainly did not intend to overinterpret findings and to convey wrong or misleading impressions. We also do acknowledge the concerns regarding the preliminary nature of the pPGR association in the breast cancer cohort. Therefore, as suggested, we removed the claims relating to the stratification of PGR+ patients based on pPGR from the abstract, toned down our conclusions and clearly mentioned the need for an independent validation at the appropriate positions in the manuscript.

We agree that our study is somewhat underpowered to detect differences in survival between the PGR+/pPGR+ and PGR+/pPGR- groups but this can, unfortunately, not be changed post hoc.

The clinical cohort available to us for testing the hypothesis derived from the p-proteome data that the phosphorylation status of PGR might further stratify PGR+ patients was not set up to ask this question, because the cohort merely represents a collection of patients who received surgery between 2004 and 2012 at the Department of Gynecology of the Klinikum rechts der Isar (as we stated in the Supplementary Information). Hence, the number of PGR+ patients (265 of 349) and the number of PGR+/pPGR- patients (41 of 265) was beyond our control. As a result, the reviewer is right in that the number of patients is insufficient to draw firm conclusions without validation in an independent cohort. The authors feel, however, that this is beyond the scope of this already very extensive study, which is why we followed the reviewer's advice to tone down that part of the manuscript.

That said, the above does not mean that our gold mine is in fact empty as suggested by the reviewer. The pPGR association was the first follow-up experiment we performed with respect to our phosphoproteomics data because we deemed it interesting. Apart from the many p-proteome-derived associations we mention throughout the manuscript, we are very confident that readers of different biological expertise will quickly find further examples in our dataset that will result in hypotheses testable using the resources available in these laboratories.

Besides this major concern with the pPGR-analysis, the other major concern which the authors have not addressed is the fact that they have computed millions of correlations without ever adjusting for multiple-testing. Once again, I did not find the author's response to be convincing, because there are many different statistical methods available to adjust for multiple testing, some very stringent and conservative (such as Bonferroni) and others which are less stringent (such as say using an $FDR < 0.05$ or even < 0.3 threshold). So, when the authors state that they did not adjust for multiple-testing because they worry about large numbers of false negatives, they should realize that this only applies to Bonferroni-type adjustments, and not to the FDR. Indeed, the FDR was invented precisely in order to have increased power (ie less false negatives) while also being able to control the number of false positives. What is not acceptable is to allow potentially large numbers of false positives to be shown on their webserver.

We acknowledge this concern and agree that controlling false discoveries is important. We therefore added FDR-adjusted P-values to the ATLANTiC web portal, which are calculated on-the-fly. For example, if a user is interested in drugs that show a significant correlation with a specific p-site, then the P-values of the correlations for all these drugs are FDR-corrected. Conversely, if a user is interested in p-sites that show a significant correlation with a specific drug, then the P-values of the correlations for all these p-sites are FDR-corrected. When the user selects more data to show, the resulting adjusted P-values will be higher. That said, we encourage users of the ATLANTiC web portal to not only rely on these FDR-adjusted P-values as the sole criterion to form a hypothesis. To facilitate the analysis, we now also show a scatter plot of FDR-adjusted P-values versus correlation coefficients, which will help users to judge or triage whether any specific association may be of further interest.

Moreover, in my comments I also asked to see a plot displaying the distribution of the fraction of drug-protein/site pairs that were evaluated over at least x number of cell-lines with x varying between 7 (the minimum number considered) and the maximum value (the number of cell-lines in the panel). This could be easily added as a SuppFig, and yet the authors have refused to show this important figure. Without this key figure, we can't be certain how comprehensive this study is, because if say 70% of the correlations were only estimated over 8 cell-lines, this would strongly undermine the claimed strengths of this study.

We apologize for not including this figure in our last revision. We have now added the plot to the manuscript (Supplementary Figure 4E).

Finally, full code is not yet available and this should be made available before reviewers assess the manuscript, not upon publication, as in my experience these promises are often not honored.

We apologize if the wording was unclear. We already uploaded the code to <https://github.com/kusterlab/ATLANTiC> more than five months ago. The link can be found in the respective section of the Supplementary Information.

Reviewer #4 (Remarks to the Author):

In the revised version of this study the authors have performed additional experiments and analyses to support the claims that the ATLANTiC tool and database can help identify pathways and targets that are unique to certain cancers. Overall, several explanations to issues raised by myself are satisfactory to support publication of this resource. There are still several issues that confound the use of this database and analysis tool and interpretation of the data in this paper. Upon addressing these issues I think this manuscript should be published.

We thank the reviewer for the positive feedback on our revision.

1) Chief among these is the lack of clarity about the relative abundance metrics used throughout the analysis of cells lines. The authors state in their supplement in response to questions by several reviewers that “if we state that a protein/phosphoprotein/p-site is ‘high’ in a certain cell line, then it means it’s abundance is higher compared to other cell lines without specifying any further metric or scale.”

Since the abundance of sites or proteins, which are hard to separate throughout the manuscript, is the core of this integrated proteomic analysis aimed at uncovering markers of disease and drug sensitivity, how can there be no metric or scale in these comparisons? This implies that there is no statistical basis behind these values. This brings into question the entire analysis and its utility if this cannot be adequately explained by the authors and presented to the readers and ultimate users of this database. There should be some sense of degree of abundance change (1.1-fold, or 100-fold?) as well as confidence in that change based on statistical measures.

The authors apologize as this did not become clear in the revision process. The abundance metrics are explained in detail in the methods section. Briefly, we use the integrated chromatography-mass spectrometric signal intensity (after normalization to make the data of different cell lines comparable to each other), which comes in values of arbitrary units (AU) and a dynamic range inherent to the MS detection system. We then perform relative comparisons using these intensities throughout the entire manuscript using fold-changes of intensity. In order to make the metrics clearer in the main text, we now state the definition of “high” and “low” at the appropriate positions in the manuscript. For example, in line 234 we extended the original statement: “We found that the sensitivity to the EGFR inhibitor Cetuximab was associated with low EPHA2 protein abundance [...]” to “We found that the sensitivity to the EGFR inhibitor Cetuximab was associated with low EPHA2 protein abundance (intensity [AU] fold-change of 342 between the lowest = $1 \times 10^{6.92}$ and the highest = $1 \times 10^{9.46}$ intensity value; [...]). Similarly, in line 345, the sentence “Our analysis suggests that high abundance of SOX10_pT240 is associated [...]” was extended to “Our analysis suggests that high abundance of SOX10_pT240 (intensity [AU] fold-change of 446 between the lowest = $1 \times 10^{6.27}$ and the highest = $1 \times 10^{8.92}$ value) is associated [...].”

2) The authors acknowledged in their response that some treatment of phosphoproteomic data, such as rolling up all phosphopeptide values into an integrated protein phosphorylation value, may be misleading. The explanation for this in the manuscript, which is that this is analogous to what is done with normal peptides for proteins, does not address the issue. The phosphorylation sites on that protein are not all placed by the same kinases, regulated by similar phosphatases, or result in similar effects on the protein or signaling. This needs to be more clearly stated in the manuscript and separated from the specific phosphorylation analyses.

The authors acknowledge that the roll-up of phosphopeptides into phosphoproteins is a strong simplification and understand that this can result in loss of information for the reasons the reviewer mentions.

However, the roll-up was necessary to be able to compare the phosphoproteomics data with public pathway databases, which contain information at the protein level only (the subject of Figure 1E). The reference to Mertins P, et al., Nature 2016 was included in order to point out that a similar concept was used before. The revised manuscript makes this clearer now by stating: “To be able to compare the phosphoproteomics data with public pathway databases, which contain information at the protein level only, we first summed up all phosphopeptide intensities for each cell line and protein group to yield phosphoprotein intensities (Supplementary Methods) similar to what was done previously (Mertins P, et al., Nature 2016). The authors point out that this is a strong simplification because phosphorylation of different sites on the same protein may be regulated by different kinases and phosphatases and can result in different effects on cellular signaling (e.g. activating versus inactivating p-sites).”

The second case that made use of the roll-up into phosphoproteins is the SMBPLSR analysis (the subject of Figures 5 and 6). Here, we decided to work with both phosphoproteins and what we term 'divergent p-sites' (i.e. p-sites that show a low correlation across cell lines with their corresponding phosphoproteins; Supplementary Methods), because it allowed us to reduce the number of variables that enter into these models but still capture the effect of single p-sites that do not parallel the abundance pattern of their corresponding phosphoprotein. The authors acknowledge that the main manuscript text may have been not entirely clear in this regard and we modified the text on a case by case basis to improve clarity.

Reviewers' Comments:

Reviewer #2:

Remarks to the Author:

This new revised version has addressed all my remaining concerns.

Reviewer #4:

Remarks to the Author:

This reviewer appreciates the revisions made to the manuscript and believes it should be published in its current form.

Detailed response to reviewer comments:

Reviewer #2 (Remarks to the Author):

This new revised version has addressed all my remaining concerns.

The authors are happy to read that the revision adequately responded to the concerns raised.

Reviewer #4 (Remarks to the Author):

This reviewer appreciates the revisions made to the manuscript and believes it should be published in its current form.

The authors are happy to read that the revision adequately responded to the concerns raised.